# Export trade structure transformation and countermeasures in the context of reverse globalization

**Xueyan Wang**[1☯], **Weidong Meng**[1☯], **Chunyang Wang**[2], **Bo Huang**[1]*, **Yuyu Li**[3]

**1** School of Economics and Business Administration, Chongqing University, Chongqing, Peoples R China,
**2** School of Economics and Management, Chongqing Jiaotong University, Chongqing, Peoples R China,
**3** School of Economics & Management, Chongqing Normal University, Chongqing, Peoples R China

☯ These authors contributed equally to this work.
* huangbo@cqu.edu.cn

**Data Availability Statement:** The data for the Reverse Globalization Index (RGI) is collected from the KOF Globalization Index published by Swiss Economic Institute. The remaining data are provided by the China Statistical Yearbook, World

## Abstract

With the development of economic globalization, the problem of unequal distribution of globalization dividends among and within countries has become increasingly serious, and reverse globalization has a great impact on the national economy and export trade. This paper uses the KOF Globalization Index and the world input-output tables in World Input-Output Database (WIOD), and empirically studies the transformation of a country's export trade and export structure in the context of reverse globalization from the perspectives of world, country, industry, subdivided manufacturing and service industry. The results show that reverse globalization has a significant non-linear negative effect on economic development and export trade. Compared with developed and European Union (EU) countries, the exports of developing and non-EU countries are more affected by reverse globalization shocks. Reverse globalization has the greatest inhibition on the secondary industry exports, followed by the tertiary industry. The suppressive effects on the exports of 12 subdivided manufacturing and 14 subdivided service in China are significantly greater than that of the United States, but most of sub-industry exports in the United States are more sensitive. Besides, China's exports of high-product-complexity industry such as metal products, medicinal chemicals, electrical and optical products and mechanical equipments are greatly affected by reverse globalization, while the exports of water transportation, construction, land transportation are relatively less restrained.

## 1. Introduction

In recent years, the economic development of the United States, Europe and other developed countries has been damaged in the unequal distribution of globalization benefits. The frequent trade wars and other issues show the decreasing economic interdependence among countries and the significant trend of reverse globalization. Reverse globalization has hindered the free flow of capital, resources, technology and other production factors, and further affects a country's economy and export trade structure. Postelnicu et al. [1] pointed out that reverse

Bank and the World Input-Output Database (WIOD). Table 1 presents the summary statistics for all of the variables.

**Funding:** This research is supported by the Cultural Experts Project (The Propaganda Department of the Central Committee of the CPC [2016]133), the Natural Science Foundation of Chongqing (No. cstc2019jcyj-msxmX0616) and the Fundamental Research Funds for the Central Universities (No.2020CDJSK02PT12). These funders have played an important role in review and editing of the manuscript.

**Competing interests:** The authors have declared that no competing interests exist.

globalization was a process of decreasing economic interdependence and integration among countries, which seriously hindered the development of world economy. The unequal distribution of globalization benefits and the urgent constraints on national sovereignty had accelerated the thinking of reverse globalization (see, [2]). Therefore, given the context of reverse globalization, we study the dynamic relationship between reverse globalization and national or industry exports and how to change and adjust a country's export trade structure, which has become an important issue for governments and scholars.

Reverse globalization is not simply deglobalization, but mainly the phenomenon of market protection in different degrees and forms after the development of economic globalization to a certain stage, which may lead to greater competition among big countries [3, 4]. Scholars have studied the impact of reverse globalization on economic growth, investment and trade. Among them, Garg and Sushil [5] analyzed the determinants of reverse globalization, such as global economy, income inequality, technological development, and other factors. Zhou et al. [6] studied the short-term and long-term impact of external reverse globalization on China's macroeconomic performance and economic growth. Xu et al. [7] found a significant negative effect of de-globalization behavior on international investment. In the development of reverse globalization, developed countries put forward more stringent trade barriers, which led to the contradiction of EU in pursuing trade interests and its bargaining power in trade negotiations, thereby restricting EU from changing the trade agreements [8], and the uncertainty of international trade environment affected the dynamic changes of a country's trade structure [9]. Using China's manufacturing exports as an example, He et al. [10] found that de-globalization leads to an increase in manufacturing export trade costs, which did not affect China's coordinated regional development, but is not conducive to manufacturing upgrading and China's economic transformation. Although the above literature has studied the impact of reverse globalization on economic growth and trade, it does not analyze the heterogeneous impact of reverse globalization on export trade in different countries, industries and subsectors.

In addition, some scholars have studied the countermeasures to deal with the impact of reverse globalization. James [11] considered that the decrease in cross-border capital flows and the slowdown in world trade growth accelerated the development of reverse globalization, while the effective reduction of trade barriers related to geographical distance and artificially high barriers can better cope with the uncertainty of international trade cooperation caused by reverse globalization [12]. He et al. [10] believed that China should improve regional transportation and communication infrastructure, reduce geographical trade costs, facilitate enterprises access to export market, maintain the export scale of existing exporters, and improve the competitiveness of enterprises to explore foreign markets, so as to reduce the impact of reverse globalization on China's manufacturing export trade. In response to the impact of reverse globalization, Garg and Sushil [5] emphasized that governments should focus on domestic manufacturing and production, strengthen policy support for local manufacturing industry, and further expand the share of sales in local markets, so as to reduce technological dependence on other countries or regions. Although the above-mentioned literature studied the countermeasures to reverse globalization, it does not give the relevant countermeasures of reverse globalization on the transformation of export trade structure of each subdivided manufacturing industry and service industry.

This paper makes three main contributions. First, most of the existing literatures discuss reverse globalization from the qualitative level, while we analyze the impact and countermeasures of reverse globalization on export trade structure from the empirical level. From the perspectives of world, country, industry, subdivided manufacturing and service industry, this paper quantitatively analyzes the dynamic impact of reverse globalization on export trade by using time series and panel data structure, and proposes some suggestions to deal with the

impact of reverse globalization. Second, we obtain valuable research results through empirical analysis. The study finds that compared to developed countries and EU countries, the exports of developing countries and non-EU countries are more affected by reverse globalization. Reverse globalization has the greatest inhibitory effect on the secondary industry exports, and the restraint effect on China's subdivided manufacturing and service industry exports is significantly greater than that of the United States. Third, through the research results, we give relevant suggestions to deal with reverse globalization. All countries need to strengthen international trade cooperation with trading partners with a more open attitude, and improve the spatial allocation efficiency of resources, technology, talents and other elements. Developing countries can take the lead in international trade cooperation on a small scale based on their geographical location, resource endowment and other advantages, and actively adjust the coordinated development of the industrial structure and export trade structure of various country, so as to alleviate the inhibitory effect of reverse globalization on export trade.

## 2. Model, variable and methodology

### 2.1. Model construction and variable selection

Sims [13] proposed the Vector Autoregression (VAR) model, which did not need to set the causal relationship between variables in advance. And Anderson and Hsiao [14] discussed the 1st difference form and the lag difference of panel data dynamic model. Panel Vector Autoregression (Panel VAR) model was firstly proposed by Holtz-Eakin et al. [15], which followed the advantages of the traditional VAR model, treated all variables as endogenous variables. This model analyzed the dynamic effects of each variable and its lagged variables on other variables, used the panel data to effectively solve the issue of individual heterogeneity, and fully considered both individual and time effects.

The form of Panel VAR model with only one variable is as follows:

$$
\begin{aligned}
Y_{it} = \quad & \alpha_0 + \alpha_1 Y_{it-1} + \cdots + \alpha_{p-1} Y_{it-p+1} + \\
& \alpha_p Y_{it-p} + f_i + d_t + \varepsilon_{it}, \quad i = 1, \cdots, N; \ t = 1, \cdots, T,
\end{aligned}
\tag{1}
$$

where $Y_{it}$ is the variable in Panel VAR model and assumes to be endogenous, $i$ is the cross-sectional dimension in panel structure, and $t$ is the time dimension. $\alpha_0$ is the intercept term and $p$ is the optimal lag order (or length) of the variable in model. $f_i$ is the fixed effect of unobserved individual heterogeneity variables, $d_t$ is the time effect of a specific cross-sectional unit, $\varepsilon_{it}$ is assumed to be a white noise error term, a stable random sequence satisfying zero mean and constant variance, and $\alpha_1, \alpha_2, \ldots, \alpha_{p-1}, \alpha_p$ is the parameter to be estimated for the model.

In the development of reverse globalization, developed countries often adopt stricter international trade barriers to protect their domestic markets [16], reduce import trade to developing countries and emerging countries, and encourage overseas multinational enterprises or factories to return home to enhance the market competitiveness of their goods [17, 18]. In consideration of national interests and geopolitics, developed countries gradually squeeze the development space of emerging market countries, hindering the free flow of global capital, resources, technology and other production factors, which eventually leads to a gradual decrease in the share of international trade and foreign direct investment flows among countries [19–21]. Therefore, reverse globalization has an inhibitory effect on a country's level of economic development and export trade.

The existing literatures failed to directly quantify the indicator of reverse globalization. In order to better evaluate the dynamic impact of reverse globalization on export trade structure, this paper uses globalization indicator for reference to measure reverse globalization. Dreher

[22] firstly proposed the globalization index covering economic, social and political aspects, and Miskiewicz and Ausloos [23] defined four macroeconomic indexes to describe the process of globalization. Samimi and Jenatabadi [24], Potrafke [25] both found that economic globalization promoted economic growth, but had a differential impact on income. Gygli et al. [26] introduced the revised version of the KOF globalization index, Dreher et al. [27] updated the degree of globalization of 195 countries or regions in the world at the economic, social and political levels since 1970, and the KOF Globalization Index had become the most widely used globalization indicator in academia. Therefore, this paper uses the KOF globalization index as the basic data, excluding the trade globalization index in the sub-dimension of economic globalization, and takes the inverse of the globalization index to obtain the Reverse Globalization Index (RGI), which is used to measure the performance of reverse globalization at the level of institutional factors such as tariffs, finance, interpersonal relationships, information, culture and politics.

Before applying the VAR model for empirical analysis, it is necessary to obtain the stationary series of each variable and the optimal lags of the model, so as to ensure the effectiveness of the model estimation results and there is no autocorrelation of the residuals. When there are $K$ variables in the model and the optimal lag order is $P$, the final estimated parameters obtained by using the VAR model are $(P * K^2 + K)$, which also includes the variance and covariance matrix of the residual [15]. Since the variables in VAR model should not be selected too much, in order to study the differential impact of a country's export structure transformation under the background of reverse globalization, this paper selects a three-variable vector, including reverse globalization index, country's economic scale, and export trade, denoted as [$RGI$, $GDP$, $EXP$]. Where $RGI$ is an indicator of reverse globalization after excluding export, which only reflects the influence of tariffs, politics and other institutional factors. $GDP$ is a measure of economic scale at the world level and in different groups of countries. $EXP$ measures the export trade at the level of world, different country categories, three major industries, and the various subdivided manufacturing and service industries.

The WIOD contains the intermediate input, intermediate consumption and final consumption of each sectors. It can clearly understand the output of various economic sectors in different countries, and how the output is allocated to other sectors for production, or to residents and society for final consumption or export to foreign countries. Since the database is only updated to the WIOD November 2016 Release, this paper mainly uses the industrial export data in world input-output tables published by WIOD in 2016 (see, [28, 29]) to study the impact of reverse globalization on export trade and export structure of industries, subdivided manufacturing and service industries for the period from 2000 to 2014. The table covers 42 countries and one region (See S1 Appendix for the specific name or location), and each country or region consists of 56 components. The study sample selects 42 countries in WIOD, which can better reflect the global production and trade patterns (see, [29]). The detailed description and source of each variable selected in this paper are shown in Table 1.

**Table 1. Variable interpretation and data source.**

| Variables | Meaning | Data Source |
|---|---|---|
| *RGI* | Reverse Globalization Index, which is the reciprocal of globalization index excluding export, mainly contains the influence of tariffs, politics and other institutional factors. | KOF Globalization Index (2000–2018) |
| *GDP* | Economic scale of the world or a country. | World Bank (2000–2018) |
| *EXP* | Exports at the level of world, different country categories, industries, and subdivided manufacturing and service industries. | World Bank (2000–2018), WIOD (2000–2014) |

*Source*: created by the authors.

## 2.2. Methodology selection

The estimation methods of Panel VAR model in existing studies include pvar estimation (see, e.g., [30–34]), pvar2 estimation (see, e.g., [35–37]) and xtvar estimation (see, e.g., [33, 38]).

The command program of pvar estimation was written by Inessa Love (University of Hawaiʻi at Mānoa), who fully considered the individual heterogeneity, the fixed effect and time effect of panel structure. The estimation of pvar2 was obtained by Ryan Decker (University of Maryland), who adapted from Inessa Love's pvar program. By fitting the lag impact of multiple panel regression of each variable on its own lag and all other variables, Helmert transformation was required to eliminate the fixed effect, and the General Method of Moments (GMM) was used for the estimation analysis. The xtvar method used a least squares dummy variable statistic to estimate VAR model, and allowed for interactions between system variables to address the endogeneity problem.

Therefore, considering the different requirements of three estimation methods for panel series and variable endogeneity test, this paper chooses the pvar2 estimation method. Due to the cross-sectional heterogeneity, the traditional maximum likelihood method can not be directly used to estimate the Panel VAR model (see, [39]). This paper uses the GMM and Monte-Carlo simulation to realize the parameter estimation.

## 3. Empirical results

### 3.1. Empirical tests and results analysis at the world level

To study the dynamic impact of reverse globalization on export trade structure transformation, we first consider the analysis at the world level. The world-level export data for this part comes from the World Bank (2000–2018). The descriptive statistical results of variables are shown in Table 2.

As can be seen from Table 2, the mean and median of *RGI* are 0.0206 and 0.0201, respectively, indicating that the degree of reverse globalization at the world level from 2000 to 2018 is relatively low, but it still shows a certain degree of reverse globalization tendency. From the Skewness and Kurtosis statistics, the conditional distributions of *GDP* and *EXP* are skewed. After log(*GDP*) and log(*EXP*) processing, the nature and correlation of the data will not be changed, but it narrows the range of variables and makes the data more stable. Although the variables log(*GDP*) and log(*EXP*) are still biased, the accompanying probability values of the two are 0.3682 and 0.3041, which are both greater than the significance level of 0.05 or even 0.1, indicating that the variables log(*GDP*) and log(*EXP*) obey the normal distribution. In addition, taking the logarithm of variables *GDP* and *EXP* can improve the effect of parameter estimation and the goodness of fit of the model (See S1 Table). Therefore, the subsequent empirical analysis directly takes the logarithm of *GDP* and *EXP*.

Using the constructed VAR model, the Augmented Dickey-Fuller (ADF) unit root test is performed on the time series of each variable, and Schwarz Information Criterion (SIC) is

**Table 2. Descriptive statistical results of variables at the world level.**

| Variables | Mean | Median | Max | Min | Std. | Skewness | Kurtosis | Prob. |
|---|---|---|---|---|---|---|---|---|
| *RGI* | 0.0206 | 0.0201 | 0.0229 | 0.0192 | 0.0013 | 0.5997 | 1.8758 | 0.3429 |
| *GDP* | 60.8592 | 63.6760 | 86.4090 | 33.4270 | 17.8701 | -0.2965 | 1.6653 | 0.4298 |
| *EXP* | 13.8414 | 15.4060 | 19.5900 | 6.2360 | 4.7700 | -0.4160 | 1.7136 | 0.3949 |
| log(*GDP*) | 4.0618 | 4.1538 | 4.4591 | 3.5094 | 0.3251 | -0.5634 | 1.8798 | 0.3682 |
| log(*EXP*) | 2.5581 | 2.7348 | 2.9750 | 1.8308 | 0.4046 | -0.7272 | 2.0554 | 0.3041 |

**Table 3. Unit root test results of the time series data (2000–2018).**

| Variables | Sequence Form | Test Condition | ADF | SIC | | |
|---|---|---|---|---|---|---|
| | | | | AIC | SC | HQ |
| *RGI* | Level | Trend and Intercept | 0.3141 | -14.9162 | -14.7678 | -14.8958 |
| | | Intercept | -3.0465** | -14.9485 | -14.8496 | -14.9349 |
| | | None | -1.5207 | -14.7931 | -14.6950 | -14.7833 |
| | 1st Difference | Trend and Intercept | -2.9817 | -14.9112 | -14.7642 | -14.8966 |
| | | Intercept | -1.9741* | -14.7618 | -14.6638 | -14.7521 |
| | | None | -1.5057 | -14.7673 | -14.7183 | -14.7625 |
| log(*GDP*) | Level | Trend and Intercept | -0.9047 | -2.8517 | -2.7033 | -2.8312 |
| | | Intercept | -1.3400 | -2.9439 | -2.8450 | -2.9302 |
| | | None | 3.3803 | -2.8792 | -2.8298 | -2.8724 |
| | 1st Difference | Trend and Intercept | -4.5394** | -3.1137 | -2.9206 | -2.8731 |
| | | Intercept | -3.2000** | -2.8829 | -2.7849 | -2.8731 |
| | | None | -1.8404* | -2.6686 | -2.6196 | -2.6637 |
| log(*EXP*) | Level | Trend and Intercept | -1.2750 | -1.2552 | -1.1068 | -1.2347 |
| | | Intercept | -1.4438 | -1.3305 | -1.1216 | -1.3169 |
| | | None | 1.8458 | -1.2640 | -1.2145 | -1.2572 |
| | 1st Difference | Trend and Intercept | -4.0647** | -1.1982 | -1.0513 | -1.1837 |
| | | Intercept | -3.7143** | -1.1867 | -1.0886 | -1.1769 |
| | | None | -3.0003*** | -1.0962 | -1.0472 | -1.0913 |

*Notes*: Three SICs are Akaike Information Criterion (AIC), Schwarz Information Criterion (SC) and Hannan-Quinn Information Criterion (HQ).

***, ** and * denote the significance levels at 1%, 5% and 10%, respectively.

selected to test *RGI*, log(*GDP*) and log(*EXP*) with trend and intercept, intercept and none under level and 1st difference series. The results are shown in Table 3.

When judged by different SIC values, theoretically the smaller the test value is, the fitting effect of each variable is better, so as to ensure the stability of variable series. Table 3 shows that *RGI* is a first-order single integer series with only intercept term. The variables log(*GDP*) and log(*GDP*) are first-order single integer series in three different test circumstances, and the fitting effect of two variables is the best in the condition of containing trend and intercept term. Therefore, the time series VAR model is analyzed by using the first-order lag stationary series for each variable.

The basic assumption of the time series VAR model is that all variables are endogenous. Therefore, it is necessary to determine the optimal lag length, and the test results are shown in Table 4. According to the significance of LR, FPE statistics and AIC, SC and HQ values, when the VAR model is in the horizontal order, only LR and SC are significant; when the VAR model is lagged by 1st order, FPE, AIC and HQ are significant. Therefore, the three-variable VAR model with lag of 1st order is finally selected.

Before performing impulse response function analysis and forecast error variance decomposition on the time series VAR model, it is necessary to conduct cointegration test on the model with the determined optimal lag period to judge whether the linear combination of stationary variable series with first-order lag has a stable equilibrium relationship, and ensure the effectiveness of VAR model estimation. The results of cointegration test are shown in Table 5.

As can be seen from the estimation results in Table 5, the time series VAR model with first-order lag is stable, because the residual series of *dRGI*, *d* log(*GDP*) and *d* log(*EXP*) are all stable, i.e., there is a cointegration relationship between the variables.

**Table 4. Results of the lagged order test at the world level.**

| Lag | LogL | LR | FPE | AIC | SC | HQ |
|---|---|---|---|---|---|---|
| 0 | 166.7847 | NA* | 6.59e-14 | -21.8380 | -21.6964* | -21.8395 |
| 1 | 177.3171 | 15.4475 | 5.58e-14* | -22.0423* | -21.4758 | -22.0483* |
| 2 | 184.9749 | 8.1683 | 8.12e-14 | -21.8633 | -20.8721 | -21.8739 |
| 3 | 195.0261 | 6.7008 | 1.28e-14 | -22.0035 | -20.5874 | -22.0186 |

*Notes*:

* indicates the optimal lagged order verified by the criterion.

LogL is the Likelihood estimation, LR is a sequential modified LR test statistic (each test at 5% level), and FPE represents the Final Prediction Error.

After determining the optimal lag length, Fig 1 describes the impulse response diagram of reverse globalization on economic scale and export trade at the world level.

Theory believes that reverse globalization is mainly affected by factors such as tariffs, international politics and social development, and it's not subject to change in economic development and export trade. As can be seen from the impulse response diagram in Fig 1, the test results also draw a similar conclusion, that is, the effects of economic scale and export trade on reverse globalization converge to zero in the next 10 period forecasts. On the contrary, reverse globalization has a significant non-linear negative effect on the world economic scale and export trade volume. Besides, from the perspective of the extent of impact, the inhibitory effect of reverse globalization on economic development is significantly smaller than its effect on export. In other words, the positive shock of reverse globalization more significantly inhibits the development of export trade. The main reason is that reverse globalization has hindered the cross-regional flow of production factors such as resources, capital, technology, labor, gradually increased international trade tariffs and barriers, and the uncertainty of international political and social development. The increase of international trade cost reduces the proportion of import and export trade and foreign direct investment flows in which countries are involved, and ultimately inhibits export trade.

## 3.2. Empirical tests and results analysis at the country level

In order to characterize the differential impact of reverse globalization on the export scale of different categories of countries, the samples, which include 42 countries in WIOD, are grouped according to whether they belong to developing countries or EU countries. In the setting of country category dummy variable, the samples belonging to developing countries are assigned a value of 0, that is, *country_category* = 0 (Cluster 1); the samples belonging to developed countries are assigned 1, i.e., *country_category* = 1 (Cluster 2). Moreover, countries that do not belong to EU are assigned a value of 0, i.e., *EU_code* = 0 (Cluster 3); the value assigning to EU countries is 1, i.e., *EU_code* = 1 (Cluster 4). Similarly, when analyzing the Panel VAR model, it is necessary to determine the stationarity of various variables. Variables *RGI*, log(*GDP*) and log(*EXP*) are tested for homogeneous unit root test, such as Levin-Lin-Chu test (LLC) (see, [40]), and heterogeneous unit root tests, which contains Im-Pesaran-Skin test

**Table 5. Results of the cointegration test at the world level.**

| Variables | | t-Statistic | Prob. |
|---|---|---|---|
| *dRGI* | *d* log(*GDP*) | -3.7975 | 0.0008 |
| *dRGI* | *d* log(*EXP*) | -2.7404 | 0.0091 |

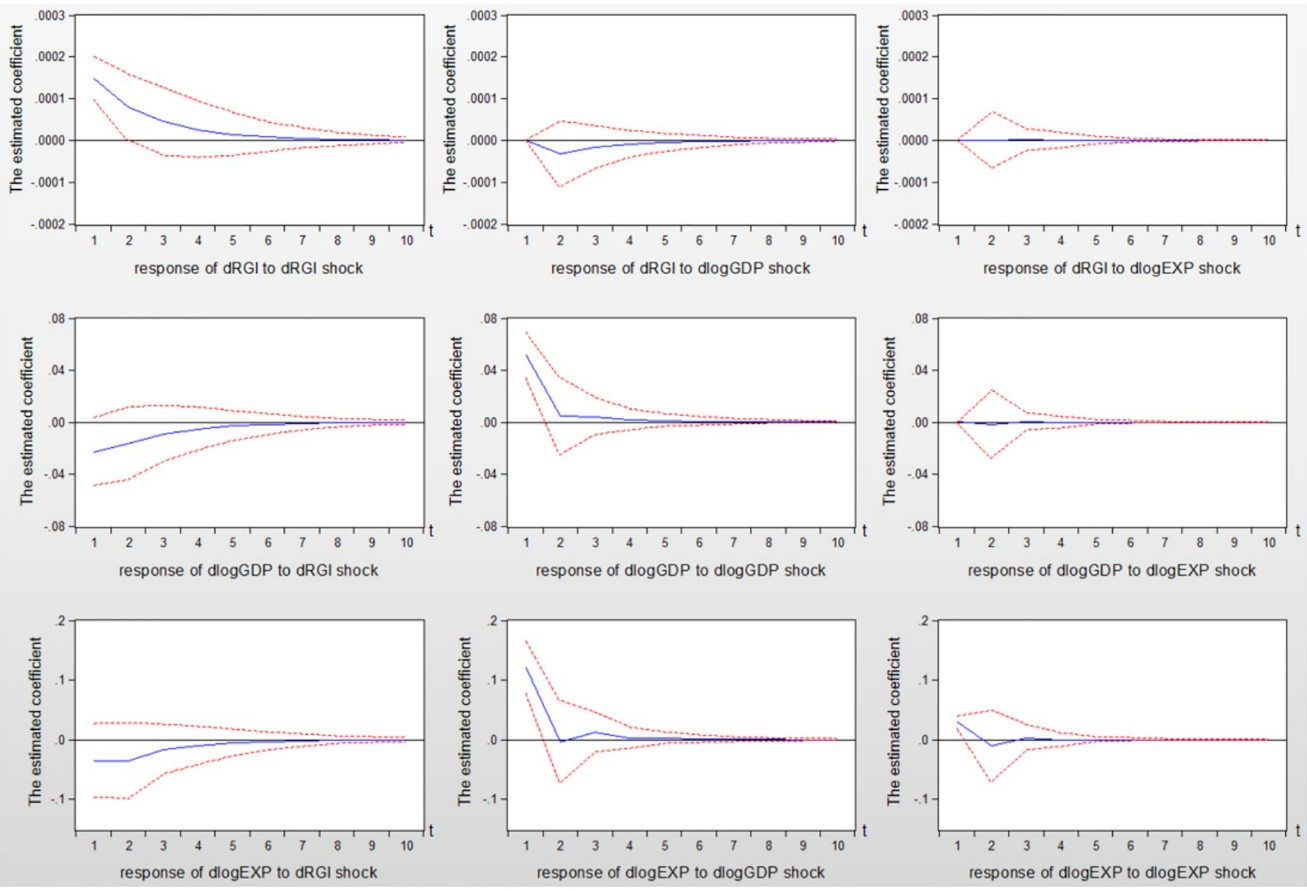

**Fig 1. Impulse response diagram of the time series VAR model.**

(IPS) (see, [41]), Fisher-ADF test (ADF) (see, [42]) and Fisher-PP test (PP) (see, [43]). Table 6 only presents the unit root test results of 1st difference series under different grouping samples.

From the unit root test results of the variables under different country groupings in Table 6, it is clear that, the probability p values of 1st difference series of $RGI$, $\log(GDP)$ and $\log(EXP)$ in all four unit root tests are less than 0.10, indicating that the 1st difference series of the three variables are stationary. Therefore, the 1st difference variables of $RGI$, $\log(GDP)$ and $\log(EXP)$ are used in the construction of the Panel VAR model at the country level.

When the 1st difference variables of $RGI$, $\log(GDP)$ and $\log(EXP)$ are stationary series, a cointegration test of each variable in the Panel VAR model is required to determine the validity of the model. The cointegration test results under different country categories are shown in Table 7.

It can be seen from the test results of different groups that the probability p-values of the cointegration test statistics for each model are less than 0.01, i.e., they are all significant at the 1% significance level, and the original hypothesis is rejected. It shows that each variable in the Panel VAR model under different grouping samples has a cointegration relationship, which means that the Panel VAR model analysis in the two cluster groups is effective.

Drawing on the pvar estimation applied by [31] and [33], it is necessary to determine the optimal lag length of the Panel VAR model (see, [44]) established under different country

**Table 6. Unit root test results of four cluster variables—1st difference series.**

| Series Tests | Cluster 1 | | | Cluster 2 | | | Cluster 3 | | | Cluster 4 | | |
|---|---|---|---|---|---|---|---|---|---|---|---|---|
| | A | B | C | A | B | C | A | B | C | A | B | C |
| *dRGI* | | | | | | | | | | | | |
| LLC | -7.062[c] | 0.000 | 15 | -3.034[b] | 0.001 | 27 | -5.253[c] | 0.000 | 14 | -4.401[b] | 0.000 | 28 |
| IPS | -2.876[a] | 0.002 | 15 | -1.349[a] | 0.089 | 27 | -2.950[a] | 0.002 | 14 | -1.753[a] | 0.040 | 28 |
| ADF | -2.972[a] | 0.000 | 15 | -8.647[b] | 0.000 | 27 | -3.134[a] | 0.001 | 14 | -9.457[b] | 0.000 | 28 |
| PP | -14.041[a] | 0.000 | 15 | -18.570[a] | 0.000 | 27 | -13.322[a] | 0.000 | 14 | -18.497[a] | 0.000 | 28 |
| $d\log(GDP)$ | | | | | | | | | | | | |
| LLC | -6.644[c] | 0.000 | 15 | -2.335[b] | 0.010 | 27 | -5.535[c] | 0.000 | 14 | -2.011[a] | 0.022 | 28 |
| IPS | -1.802[b] | 0.036 | 15 | -3.033[a] | 0.001 | 27 | -2.366[a] | 0.009 | 14 | -2.839[b] | 0.002 | 28 |
| ADF | -6.479[b] | 0.000 | 15 | -2.607[a] | 0.005 | 27 | -2.323[a] | 0.010 | 14 | -9.061[b] | 0.000 | 28 |
| PP | -4.297[a] | 0.000 | 15 | -10.104[a] | 0.000 | 27 | -7.330[a] | 0.000 | 14 | -4.867[a] | 0.000 | 28 |
| $d\log(EXP)$ | | | | | | | | | | | | |
| LLC | -2.158[a] | 0.015 | 15 | -5.179[a] | 0.000 | 27 | -1.287[b] | 0.099 | 14 | -2.730[a] | 0.003 | 28 |
| IPS | -2.695[a] | 0.004 | 15 | -6.775[a] | 0.000 | 27 | -1.795[a] | 0.036 | 14 | -4.029[a] | 0.000 | 28 |
| ADF | -2.841[a] | 0.002 | 15 | -6.617[a] | 0.000 | 27 | -1.648[a] | 0.050 | 14 | -3.985[a] | 0.000 | 28 |
| PP | -7.852[a] | 0.000 | 15 | -17.715[a] | 0.000 | 27 | -12.240[a] | 0.000 | 14 | -14.029[a] | 0.000 | 28 |

*Notes*: A is the value of each test statistic, B is the p value corresponding to each test, and C represents the number of cross sections. The letters *a*, *b* and *c* respectively express the three circumstances where the variable contains trend and intercept item, only intercept item and none, and indicate the situation in which the test statistic value is significant.

categories to ensure the statistical credibility of the model. When selecting the optimal lag order, three information criterias of MBIC, MAIC, and MQIC are generally used for judgment, and the minimum value of information criterion is the optimal lag order. When the three results are inconsistent, it is generally considered that the MBIC/MQIC standard is better than MAIC. The test results are shown in Table 8.

It can be seen from Table 8, when the Panel VAR model is lagged by 1st-order, the p values of J statistic are less than 0.05, indicating that the optimal lag length of the model is 1. In addition, the lagged first-order values of MBIC, MAIC, and MQIC are the smallest, so it is reasonable to choose the lagged first-order of the Panel VAR model to study the dynamic impact of reverse globalization on economic scale and export trade in different country categories.

After eliminating the fixed effect and country-year dummy variable, this paper considers the robustness of GMM, combines with 2000 reps of Monte-Carlo simulation to estimate the

**Table 7. Results of the cointegration test at the country level.**

| Variables | MDFt | | DFt | | UMDFt | | UDFt | |
|---|---|---|---|---|---|---|---|---|
| | Statistic | p-value | Statistic | p-value | Statistic | p-value | Statistic | p-value |
| Panel A. Cluster 1 (country_category = 0) $dRGI$, $d\log(GDP)$, $d\log(EXP)$ | -9.446 | 0.000 | -14.556 | 0.000 | -16.310 | 0.000 | -16.015 | 0.000 |
| Panel B. Cluster 2 (country_category = 1) $dRGI$, $d\log(GDP)$, $d\log(EXP)$ | -19.413 | 0.000 | -16.650 | 0.000 | -21.779 | 0.000 | -16.939 | 0.000 |
| Panel C. Cluster 3 (EU_code = 0) $dRGI$, $d\log(GDP)$, $d\log(EXP)$ | -8.994 | 0.000 | -13.015 | 0.000 | -13.835 | 0.000 | -13.962 | 0.000 |
| Panel d. Cluster 4 (EU_code = 1) $dRGI$, $d\log(GDP)$, $d\log(EXP)$ | -15.306 | 0.000 | -16.552 | 0.000 | -21.976 | 0.000 | -17.676 | 0.000 |

*Notes*: MDFt: Modified Dickey-Fuller t. DFt: Dickey-Fuller t. UMDFt: Unadjusted modified Dickey-Fuller t. UDFt: Unadjusted Dickey-Fuller t.

**Table 8. Optimal lag length tests of the Panel VAR model under different country categories.**

| Lag | CD | J p-value | MBIC | MAIC | MQIC |
|---|---|---|---|---|---|
| Panel A. Cluster 1 | | | | | |
| 1 | 43.702 | 0.022 | -98.669* | -10.298 | -46.078* |
| 2 | 22.417 | 0.214 | -72.497 | -13.583* | -37.437 |
| 3 | 7.404 | 0.595 | -40.053 | -10.596 | -22.522 |
| Panel B. Cluster 2 | | | | | |
| 1 | 74.078 | 2.88e-06 | -84.163* | -20.078 | -21.409* |
| 2 | 66.055 | 2.08e-07 | -39.439 | -30.055* | -2.397 |
| 3 | 8.857 | 0.451 | -43.890 | -9.143 | -20.972 |
| Panel C. Cluster 3 | | | | | |
| 1 | 45.208 | 0.015 | -95.301* | -8.792 | -43.862* |
| 2 | 17.451 | 0.492 | -76.221 | -18.549* | -41.928 |
| 3 | 10.922 | 0.281 | -35.914 | -7.078 | -18.768 |
| Panel D. Cluster 4 | | | | | |
| 1 | 108.574 | 9.66e-12 | -50.650* | -54.573* | -12.752* |
| 2 | 62.141 | 9.18e-07 | -44.008 | -26.141 | -1.740 |
| 3 | 21.718 | 0.010 | -31.357 | -3.718 | -10.223 |

*Notes*: J represents the Jonhamson Test. Three information criterias are MMSC-Bayesian Information Criterion (MBIC), MMSC-Akaike Information Criterion (MAIC), and MMSC-Hannan and Quinn Information Criterion (MQIC).

* indicates the minimum value under MBIC, MAIC and MQIC, and the value corresponding to the order is the optimal lag order.

model parameters, and obtains the impulse response function diagram and the prediction error variance decomposition results of Panel VAR model under different country categories by pvar2 estimation.

Figs 2 and 3 describe the impulse response diagram from developing and developed countries. Like the analysis at the world level, the impact of a country's economic scale and export trade under different country groupings on reverse globalization is close to 0. In addition, a standard deviation shock of reverse globalization has a significant negative effect on a country's economic scale and export, and has a greater effect on the country's export trade. Comparing the results of Cluster 1 and 2, it is found that the effect of reverse globalization on developing countries is slightly greater than that of developed countries. The possible reason is that, with the increasing strength of trade protectionism in developed countries such as Europe and the United States, countries rely less on cross-regional cooperation in trade in goods and services, encourage multinational enterprises and factories to return home to enhance the competitiveness of their own products, and their national export trade is less affected by reverse globalization. However, with the increasing uncertainty of international trade, reverse globalization has seriously hindered the opportunities for developing countries to participate in international economic and trade cooperation, restricted the spatial flow of various resources, capital, technology and other factors, and significantly increased trade costs, which in turn has a greater impact on the exports of developing countries.

Figs 4 and 5 describe the impulse response diagram of non-EU and EU countries, respectively. The figures show that a country's economic scale and export trade have no effect on reverse globalization. Comparing the results of Cluster 3 and 4, it is found that the inhibitory effect of reverse globalization on the export of non-EU countries is higher than that of EU countries. The possible reason is that the EU, as the economic and political community of European countries, has a certain economic strength in its entire common market to resist the

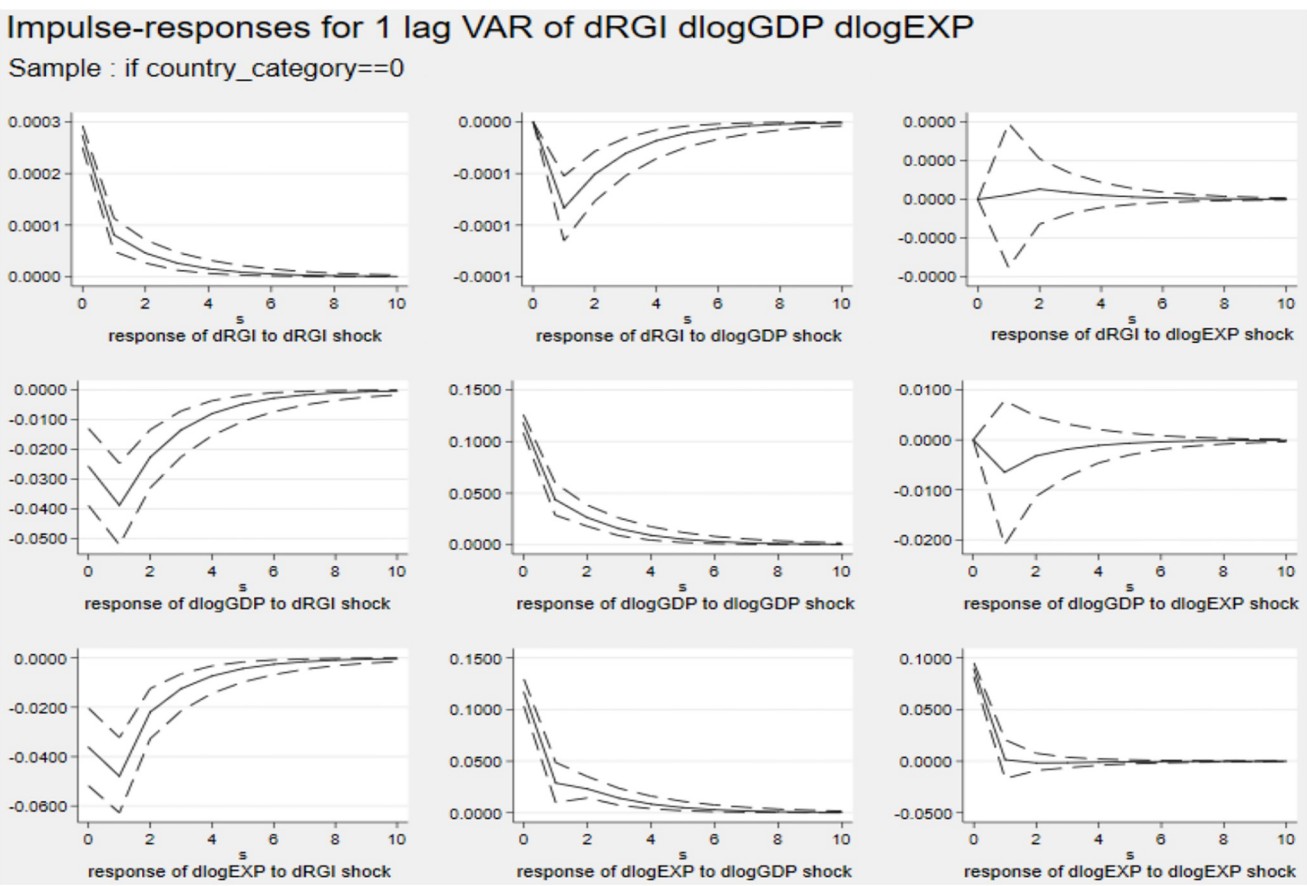

**Fig 2. Impulse response diagram in Cluster 1.** *Notes*: Errors are 5% on each side generated by Monte-Carlo with 2000 reps.

reverse globalization shocks. However, the participation of non-EU countries in multi-bilateral international trade cooperation is more vulnerable to factors such as tariffs, finance and international political instability. Since it is more costly for non-EU countries to participate in trade cooperation, the impact of reverse globalization on export trade is greater.

The forecast error variance decomposition is used to analyse the relative contribution rate of reverse globalization to economic scale and export fluctuation for different country categories. The result of period 10 is given in Table 9.

In Cluster 1, reverse globalization has a strong ability to explain export changes, reaching 15.69% in period 10, indicating that 15.69% of developing countries' export trade changes can be explained by reverse globalization. While in Cluster 2, the explanation is only 2.86%. In Cluster 3 and 4, the explanation rate are 4.71% and 14.26%, respectively. These analyses are consistent with the impulse response results.

Table 10 reports the Granger causality test results of the Panel VAR model at the country level. Consistent with economic theory, there is a significant effect of reverse globalization on a country's economy scale and export trade at a significance level of 1%.

### 3.3. Empirical tests and results analysis at the industry level

In order to analyse the dynamic differential impact of reverse globalization on a country's industrial export, the samples of 42 countries in WIOD are grouped by three major industries.

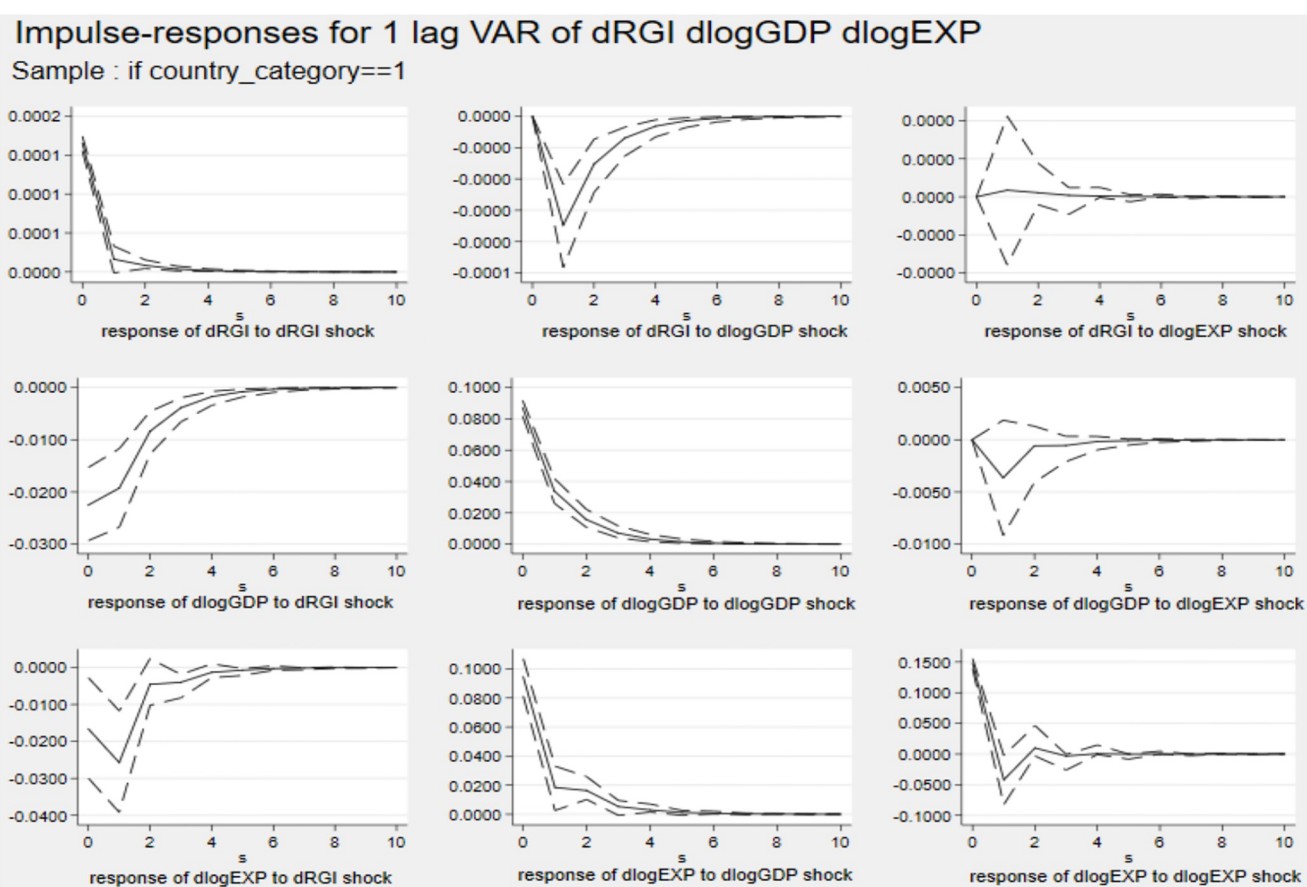

**Fig 3. Impulse response diagram in Cluster 2.** *Notes*: Errors are 5% on each side generated by Monte-Carlo with 2000 reps.

In the setting of industrial category variable, part A in the International Standard Industrial Classification Revision 4 (ISIC Rev.4) is the primary industry and is assigned a value of 1, namely *industry_category* = 1 (Cluster 5); part B-F in the industry classification table is the secondary industry and the value is 2, that is, *industry_category* = 2 (Cluster 6); part G-U is the tertiary industry and is assigned a value of 3, i.e., *industry_category* = 3 (Cluster 7).

With reference to the analysis at the world and country level, under different industry groups, the 1st difference series of *RGI*, log(*GDP*), log(*EXP*) and the Panel VAR model with optimal lag length of 1 are used for impulse response and variance decomposition to study the heterogeneous impact of reverse globalization on export. The results of pvar2 estimation under different industry groups are presented in Table 11.

In Cluster 5, 6 and 7, there is a significant inhibitory effect of reverse globalization on industrial exports. In Cluster 5 and 6, the negative effect of reverse globalization on is significant at a significance level of 1%, while in Cluster 7, the inhibitory effect is significant at a significance level of 5%. In terms of the magnitude of the coefficient, reverse globalization has the largest inhibitory effect on exports in the secondary industry, followed by the tertiary industry, and the smallest is the primary industry. This is principally because the secondary industrial exports are the main components of international trade, the manufacturing and construction industrial exports need a large number of production factors such as resources, capital and technology, reverse globalization increases the flow cost of various factors of production.

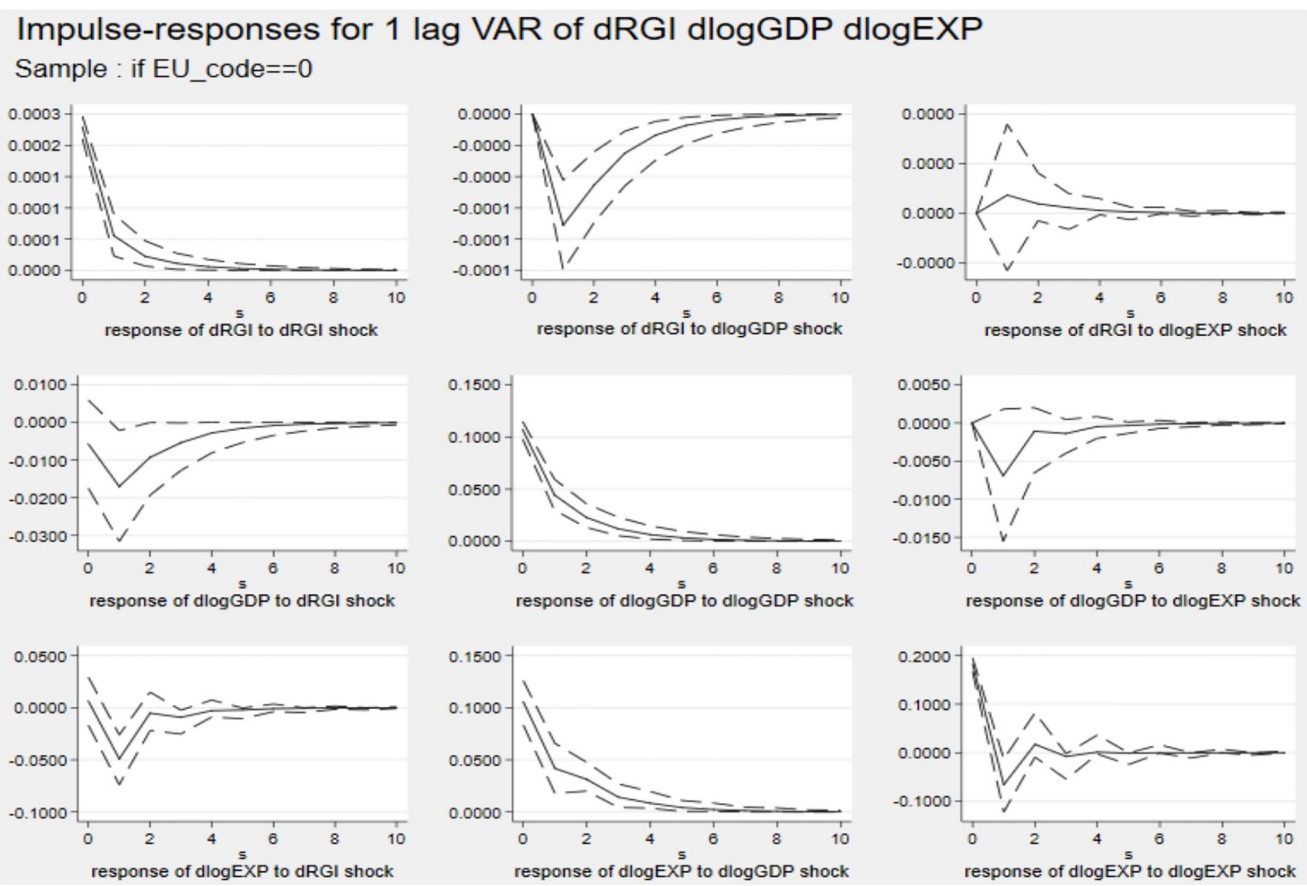

**Fig 4. Impulse response diagram in Cluster 3.** *Notes*: Errors are 5% on each side generated by Monte-Carlo with 2000 reps.

Industries with high product complexity often require more stringent technology and resource constraints, and countries have longer cycles of industrial restructuring, which eventually leads to the greater impact of reverse globalization on secondary industry exports.

Figs 6–8 describe the dynamic impulse response diagrams under different industrial groups generated by Monte-Carlo simulation with 2000 reps, respectively.

In the above three figures, after giving a standard deviation shock to reverse globalization at the initial stage, the effects on exports of three major industries are all negative. The first period has the largest inhibitory effect on those industries' export. After the second period, the negative impact on industry export gradually weakens and tends to be stable. Furthermore, there are significant differences in the negative effects of reverse globalization on exports of various industries. Among them, the effects of reverse globalization on exports of the primary industry and the tertiary industry are not much different, but it has the largest impact on exports of the secondary industry.

Table 12 gives the relative contribution rate of reverse globalization to export fluctuation under different industry groups. The results find that the explanatory power of reverse globalization on exports of the three major industries is 7.71%, 18.41% and 16.62%, respectively, i.e., the impact of reverse globalization has a greater inhibitory effect on the secondary industry exports, which is consistent with the pvar2 estimation and impulse response results.

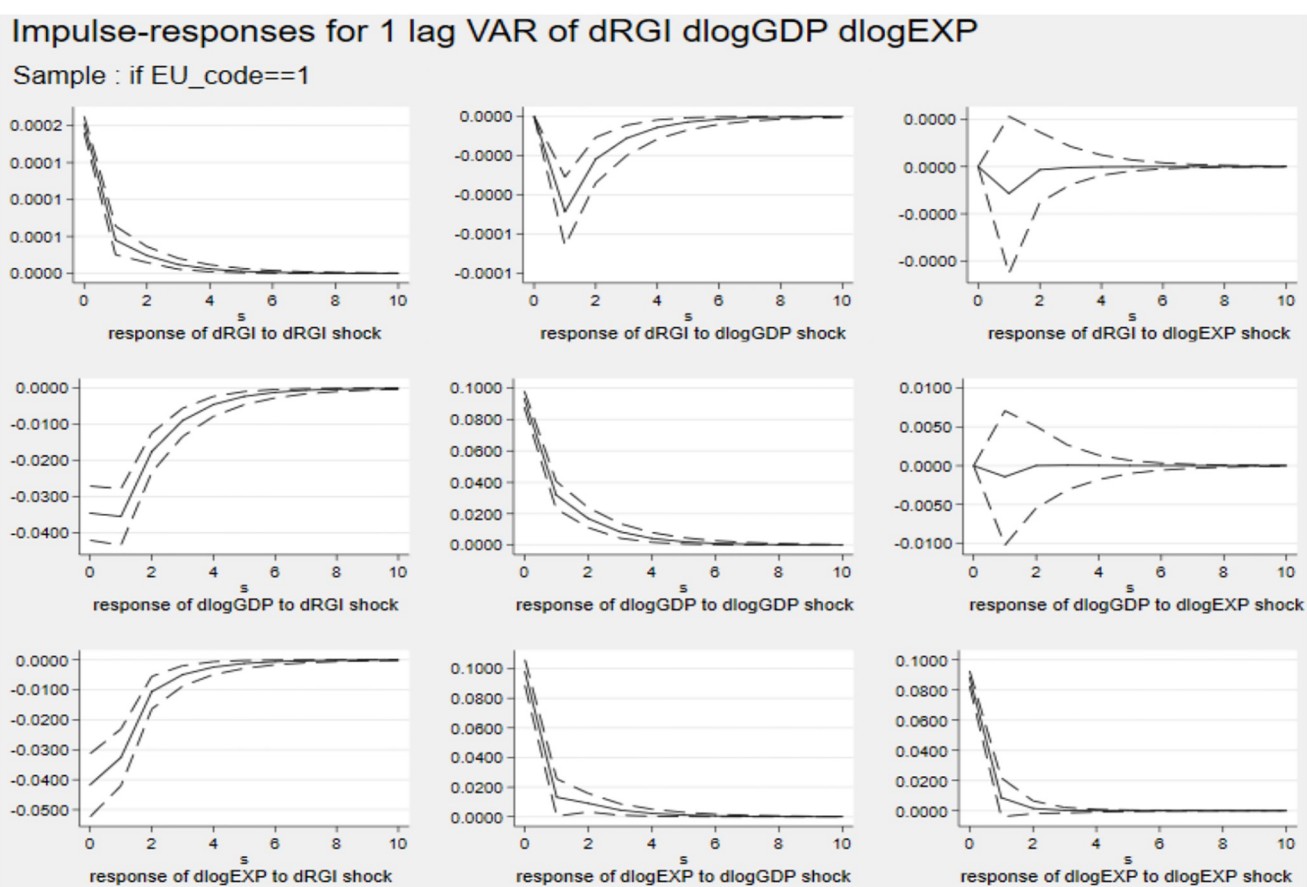

**Fig 5. Impulse response diagram in Cluster 4.** *Notes*: Errors are 5% on each side generated by Monte-Carlo with 2000 reps.

In the following, taking the industries of China and the United States as examples, Fig 9 describes the dynamic impact of reverse globalization on industrial exports bewteen China and the United States.

It turns out that, with the exception of China's primary industry exports, reverse globalization has a significant negative effect on the industrial exports of China and the United States. The inhibitory effect on China's industrial export reaches the maximum in the second period, while its effect on the United States reaches the maximum in the first period, that is, the United States industrial exports are more sensitive to the effects of reverse globalization. In terms of

**Table 9. Variance decomposition results under different country categories (unit: %).**

| Cluster | *dRGI* | *d* log(*GDP*) | *d* log(*EXP*) | Cluster | *dRGI* | *d* log(*GDP*) | *d* log(*EXP*) |
|---|---|---|---|---|---|---|---|
| Panel A. Cluster 1 | | | | Panel B. Cluster 2 | | | |
| *dRGI* | 88.48 | 11.46 | 0.05 | *dRGI* | 94.83 | 5.15 | 0.02 |
| *d* log(*GDP*) | 14.85 | 84.86 | 0.29 | *d* log(*GDP*) | 9.66 | 90.20 | 0.14 |
| *d* log(*EXP*) | 15.69 | 28.91 | 55.40 | *d* log(*EXP*) | 2.86 | 28.00 | 69.15 |
| Panel C. Cluster 3 | | | | Panel D. Cluster 4 | | | |
| *dRGI* | 87.47 | 12.41 | 0.12 | *dRGI* | 93.49 | 6.44 | 0.07 |
| *d* log(*GDP*) | 3.05 | 96.59 | 0.35 | *d* log(*GDP*) | 22.13 | 77.85 | 0.02 |
| *d* log(*EXP*) | 4.71 | 25.99 | 69.30 | *d* log(*EXP*) | 14.26 | 37.97 | 47.76 |

**Table 10. Granger causality test results under different country categories (some main results).**

| Equation | Excluded | chi2 | df | Prob > chi2 |
|---|---|---|---|---|
| Panel A. Cluster 1 | | | | |
| $d RGI$ | $d \log(GDP)$ | 0.000 | 1 | 0.987 |
| | $d \log(EXP)$ | 0.499 | 1 | 0.480 |
| $d \log(GDP)$ | $d RGI$ | 28.501 | 1 | 0.000 |
| $d \log(EXP)$ | $d RGI$ | 30.230 | 1 | 0.000 |
| Panel B. Cluster 2 | | | | |
| $d RGI$ | $d \log(GDP)$ | 0.117 | 1 | 0.732 |
| | $d \log(EXP)$ | 1.831 | 1 | 0.176 |
| $d \log(GDP)$ | $d RGI$ | 10.027 | 1 | 0.002 |
| $d \log(EXP)$ | $d RGI$ | 7.504 | 1 | 0.006 |
| Panel C. Cluster 3 | | | | |
| $d RGI$ | $d \log(GDP)$ | 8.288 | 1 | 0.004 |
| | $d \log(EXP)$ | 3.197 | 1 | 0.069 |
| $d \log(GDP)$ | $d RGI$ | 20.483 | 1 | 0.000 |
| $d \log(EXP)$ | $d RGI$ | 26.476 | 1 | 0.000 |
| Panel D. Cluster 4 | | | | |
| $d RGI$ | $d \log(GDP)$ | 2.295 | 1 | 0.130 |
| | $d \log(EXP)$ | 2.406 | 1 | 0.121 |
| $d \log(GDP)$ | $d RGI$ | 10.248 | 1 | 0.001 |
| $d \log(EXP)$ | $d RGI$ | 8.704 | 1 | 0.003 |

the degree of inhibition, there is little difference in the effect of reverse globalization on the primary industry export, while the inhibition on the exports of China's secondary industry and tertiary industry is significantly greater than that of the United States. This is consistent with the analysis results under different industry groups. The main reason is that the United States implements trade protectionism and implements higher tariff and non-tariff barriers to the outside world, which makes the trade partner countries reduce their trade with the export of relevant industries in the United States, namely, the United States industrial exports are more sensitive to the impact of reverse globalization. However, As the beneficiary and promoter of globalization, China's industrial export response is slightly slower than that of the impact of reverse globalization, and reaches the maximum restraint effect after a period of lag. Therefore, China and the United States need to carry out foreign trade cooperation with a more open attitude, actively adjust the industrial structure and export trade structure, and improve the efficiency of resource allocation.

## 3.4. Empirical tests and results analysis at the subdivided manufacturing and service industry level

To further study the differential impact of reverse globalization on the exports in different industries, the subdivided manufacturing and service industry of major exporting countries in WIOD are analysed. Referring to the division of China's 14 subdivided manufacturing and 17 service industry in [45], combined with the ISIC Rev.4, the manufacturing and service industries are subdivided as shown in Table 13.

According to the input-output data of 42 countries in WIOD, the top three exports and the proportions between China and the United States are aggregated by the subdivided manufacturing and service industry from Table 13, as shown in Figs 10 and 11.

**Table 11. Results of pvar2 estimation under different industry groups (some main results).**

| Variables | | Coef. | Std.Err. | z | p > \| z \| | [95% Conf.Interval] | |
|---|---|---|---|---|---|---|---|
| Panel E. Cluster 5 | | | | | | | |
| dRGI | dRGI L1. | 0.4015 | 0.1377 | 2.92 | 0.004 | 0.1317 | 0.6714 |
| | d log(GDP) L1. | -0.0006 | 0.0002 | -2.59 | 0.010 | -0.0010 | -0.0001 |
| | d log(EXP) L1. | 0.0006 | 0.0001 | 3.74 | 0.000 | 0.0003 | 0.0008 |
| d log(GDP) | dRGI L1. | -458.0348 | 109.7610 | -4.17 | 0.000 | -673.1625 | -242.9071 |
| | d log(GDP) L1. | 0.0627 | 0.1637 | 0.38 | 0.702 | -0.2581 | 0.3836 |
| | d log(EXP) L1. | -0.1542 | 0.0944 | -1.63 | 0.102 | -0.3392 | 0.0308 |
| d log(EXP) | dRGI L1. | -48.2893 | 127.0622 | -3.84 | 0.000 | -737.3267 | -239.2520 |
| | d log(GDP) L1. | -0.2450 | 0.2268 | -1.08 | 0.280 | -0.6896 | 0.1996 |
| | d log(EXP) L1. | -0.0820 | 0.2575 | -0.52 | 0.630 | -0.6907 | 0.2267 |
| Panel F. Cluster 6 | | | | | | | |
| dRGI | dRGI L1. | 0.4533 | 0.1416 | 3.20 | 0.001 | 0.1759 | 0.7308 |
| | d log(GDP) L1. | 0.0004 | 0.0004 | 1.03 | 0.304 | -0.0004 | 0.0012 |
| | d log(EXP) L1. | -0.0005 | 0.0003 | -1.76 | 0.079 | -0.0010 | 0.0001 |
| d log(GDP) | dRGI L1. | -417.7743 | 99.8554 | -4.18 | 0.000 | -613.4873 | -222.0614 |
| | d log(GDP) L1. | -0.3542 | 0.3339 | -1.06 | 0.289 | -1.0087 | 0.3002 |
| | d log(EXP) L1. | 0.3060 | 0.2317 | 1.32 | 0.187 | -0.1481 | 0.7601 |
| d log(EXP) | dRGI L1. | -534.2489 | 133.5197 | -4.00 | 0.000 | -795.9427 | -272.5552 |
| | d log(GDP) L1. | -0.6208 | 0.4905 | -1.27 | 0.206 | -1.5822 | 0.3405 |
| | d log(EXP) L1. | 0.4795 | 0.3496 | 1.37 | 0.170 | -0.2057 | 1.1648 |
| Panel G. Cluster 7 | | | | | | | |
| dRGI | dRGI L1. | 0.3038 | 0.1615 | 1.88 | 0.060 | -0.0128 | 0.6205 |
| | d log(GDP) L1. | 0.0007 | 0.0006 | 1.08 | 0.279 | -0.0006 | 0.0019 |
| | d log(EXP) L1. | -0.0008 | 0.0007 | -1.19 | 0.235 | -0.0021 | 0.0005 |
| d log(GDP) | dRGI L1. | -361.3138 | 142.4859 | -2.54 | 0.011 | -640.5810 | -82.0466 |
| | d log(GDP) L1. | 0.7948 | 0.5844 | 1.36 | 0.174 | -0.3505 | 1.9402 |
| | d log(EXP) L1. | -0.8516 | 0.6525 | -1.31 | 0.192 | -2.1304 | 0.4272 |
| d log(EXP) | dRGI L1. | -351.9323 | 138.0043 | -2.55 | 0.011 | -622.4157 | -81.4489 |
| | d log(GDP) L1. | 0.6403 | 0.5460 | 1.17 | 0.241 | -0.4299 | 1.7105 |
| | d log(EXP) L1. | -0.6415 | 0.6141 | -1.04 | 0.296 | -1.8450 | 0.5620 |

In Fig 10, manufacturing exports are mainly concentrated in metals, electrical and optical products, transportation equipments. Except for paper, refined petroleum and transportation equipments, China's exports proportion of sub-manufacturing industries is greater than that of the United States. In Fig 11, service exports are mainly concentrated in public management, retail trade, and real estate industries. With the exception of water transportation, electricity and gas, construction and education, the proportion of exports of other service industries in the United States is significantly greater than that of China. Judging from total values, the United States has an obvious export advantage in service industry, while China has a small manufacturing export advantage.

According to the industry classification in ISIC Rev.4, China (CHN) and the United States (USA) are selectedi to analyse the dynamic impact of reverse globalization on exports of various sub-industry. Figs 12 and 13 depict the impulse response diagram of reverse globalization to sub-manufacturing exports of China and the United States.

As can be seen from Fig 12, after giving a standard deviation shock to reverse globalization in the current period, the effects on China's exports of 12 sub-manufacturing industry are

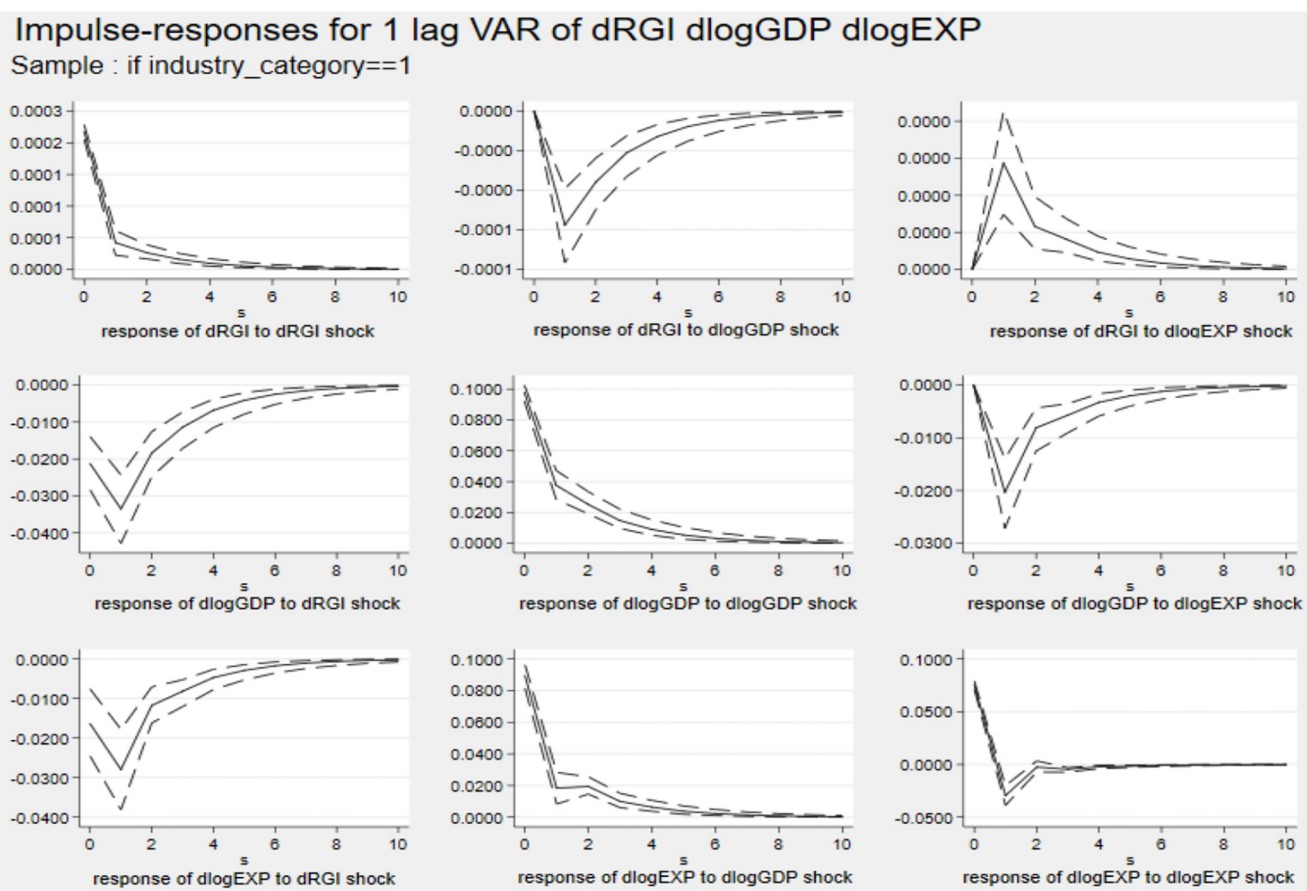

**Fig 6. Impulse response diagram in Cluster 5.** *Notes*: Errors are 5% on each side generated by Monte-Carlo with 2000 reps.

mostly negative in the first four periods. And some sub-manufacturing industry shows smaller positive effects gradually weakening and leveling off after fourth period, indicating that reverse globalization is closely related to China's various sub-manufacturing export trade. Reverse globalization has a greater inhibitory effect on China's sub-manufacturing industry numbered 05, 09 and 12, followed by industry numbered 06 and 11, and the least impact on industry numbered 04 and 07. Besides, the negative effects of reverse globalization on sub-manufacturing industry numbered 05, 07, 08, 10 and 12 reach its maximum in the first period, that is, the exports of such sub-manufacturing industry are more sensitive to the impact of reverse globalization. The negative effects on industry numbered 01, 02, 04, 06, 09 and 11 reach the maximum in the second period, while the impact on industry numberd 03 fluctuates greatly.

Similarly, in Fig 13, when reverse globalization is given a standard deviation shock in the current period, the effects on 12 subdivided manufacturing industries in the United States are negative in the first three periods, while the promotion effects gradually weaken and tend to zero after third period, indicating that reverse globalization is closely related to the development of the United States in various subdivisions. Reverse globalization has the greatest inhibitory effect on the United States sub-manufacturing industry numbered 05, 09, 11 and 12, while it has less impact on industry numbered 01, 02, and 03. The negative impact of reverse globalization on exports of the United States sub-manufacturing industry is all reached the maximum in the first period.

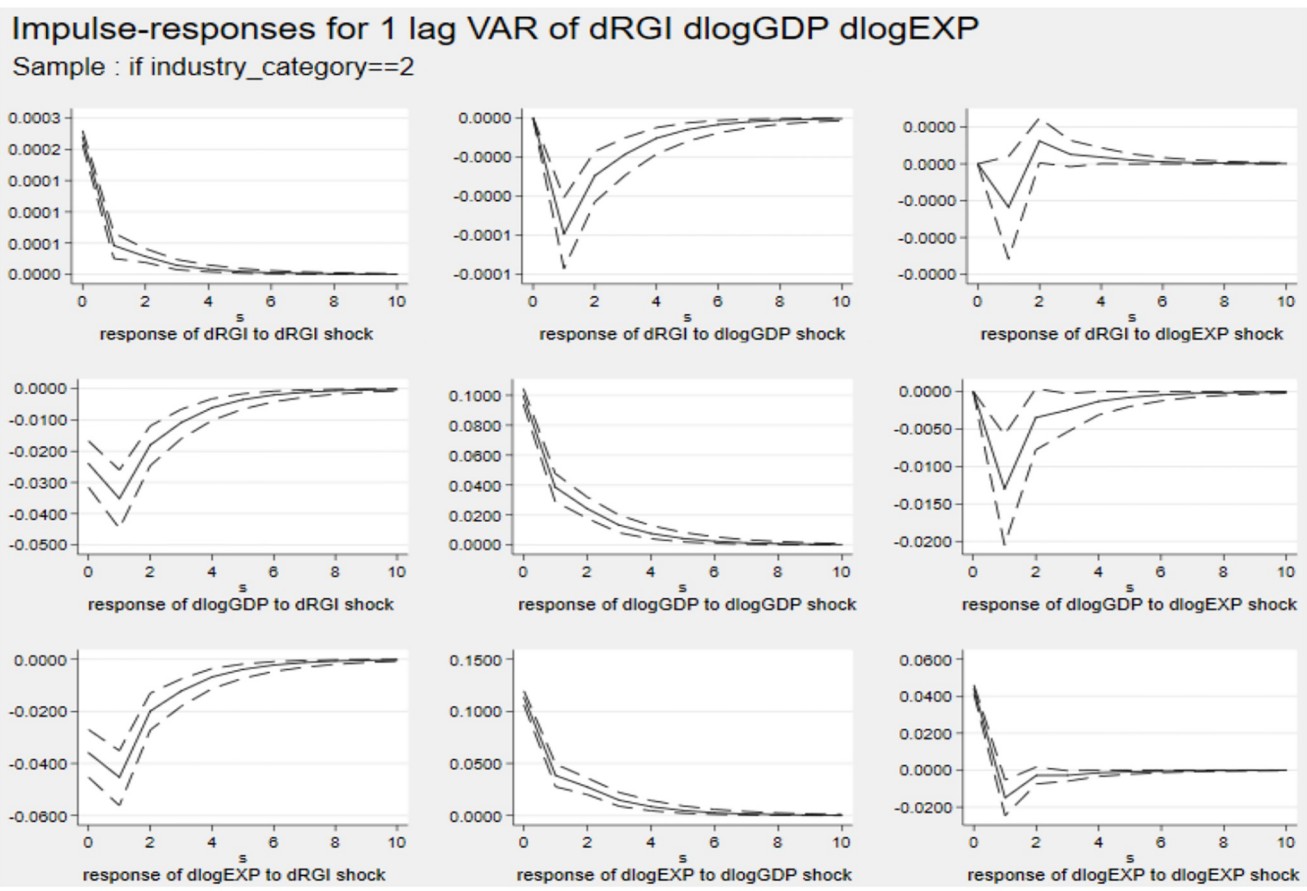

**Fig 7. Impulse response diagram in Cluster 6.** *Notes*: Errors are 5% on each side generated by Monte-Carlo with 2000 reps.

Combining Figs 12 and 13, there is little difference in the inhibition of reverse globalization on manufacturing exports between China and the United States for the subdivision of industry numbered 04, 05, 07, 09, 10, 11 and 12. In industry numbered 02, 03 and 08, the scale of China's sub-manufacturing industry exports is much larger than that of the United States, but the inhibitory effect on China's manufacturing is significantly greater than that of the United States. In industry numbered 01 and 06, the size of China's manufacturing exports is not much different from the United States, while reverse globalization has a greater impact on China's sub-manufacturing industry numberd 01 and 06. The possible reason for this is that, China's vast territory, large population, and abundant total resources make China as a major manufacturing exporter. In the development of reverse globalization, developed countries such as the United States encourage the return of multinational manufacturing enterprises and factories, which increases the unemployment pressure of Chinese residents and reduces China's effective manufacturing output. The share of China's manufacturing exports in international trade has decreased, which hinders the development of China's manufacturing industry. As a result, the inhibitory effect of reverse globalization on China's various sub-manufacturing exports is slightly greater than that of the United States.

Figs 14 and 15 depict the impulse response diagram of the impact of reverse globalization on 14 sub-service industry exports in China and the United States.

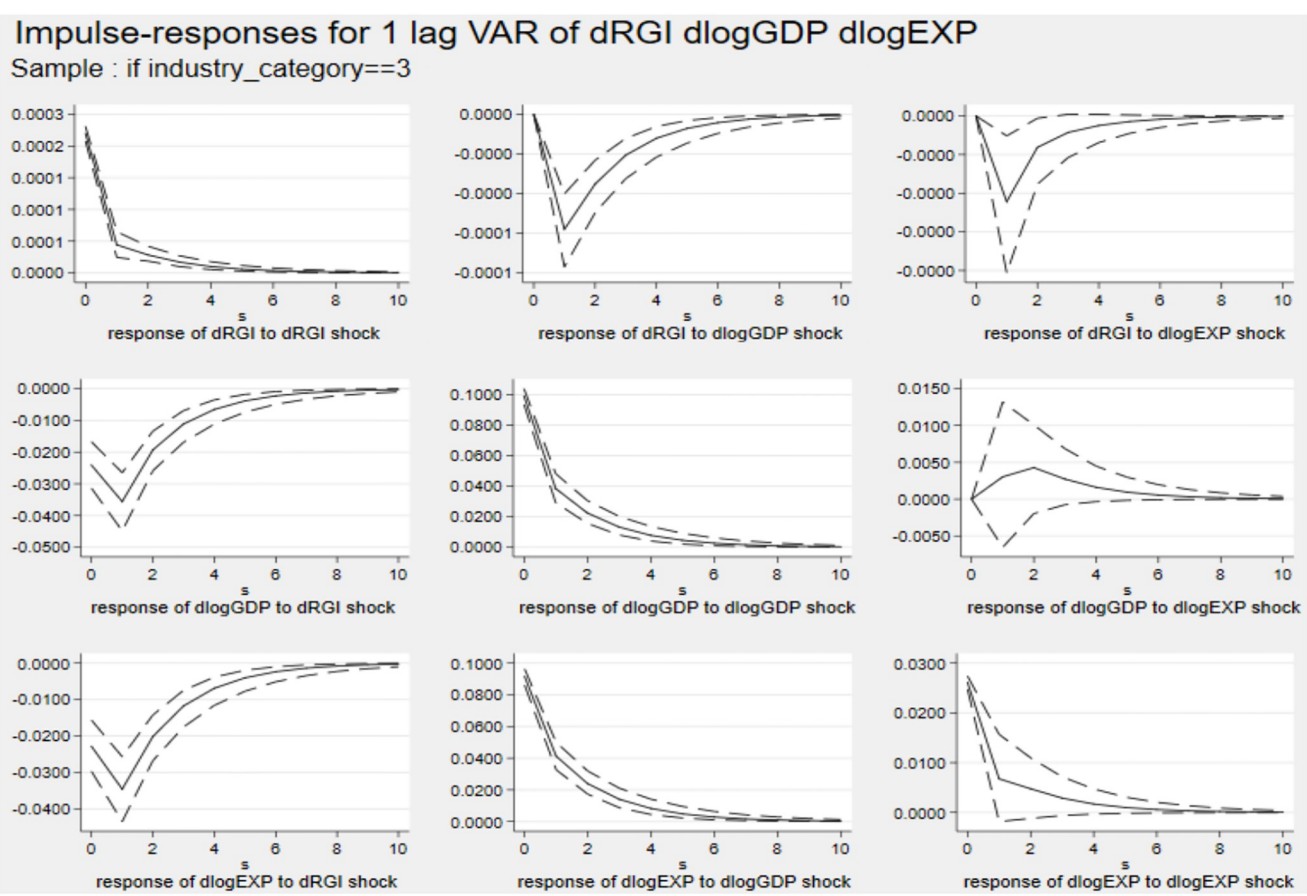

**Fig 8. Impulse response diagram in Cluster 7.** *Notes*: Errors are 5% on each side generated by Monte-Carlo with 2000 reps.

Consistent with the analysis in Figs 12 and 13, it can be seen from Figs 14 and 15 that after giving a shock to reverse globalization, the effects on 14 sub-service industry are negative in the first three periods, and gradually level off after third period. As a whole, the inhibitory effect of reverse globalization on the exports of China's sub-service industry is significantly greater than that of the United States. Under the influence of reverse globalization, most of China's sub-service industry have the greatest negative effect in the second period, while the United States reach the strongest inhibitory effect in the first period. The result shows that the various sub-service industry in the United States is more sensitive to reverse globalization shocks, but the degree of impact is less than that of China. The main reason is that the United States has obvious advantages in services exports. Its education, scientific and technological research, finance, insurance, communications and other industries have become the main

**Table 12. Variance decomposition results under different industry groups (unit: %).**

| Variables | Panel E: Cluster 5 | | | Panel F: Cluster 6 | | | Panel G: Cluster 7 | | |
|---|---|---|---|---|---|---|---|---|---|
| | *dRGI* | *d* log(*GDP*) | *d* log(*EXP*) | *dRGI* | *d* log(*GDP*) | *d* log(*EXP*) | *dRGI* | *d* log(*GDP*) | *d* log(*EXP*) |
| *dRGI* | 88.78 | 9.36 | 1.86 | 91.00 | 8.66 | 0.33 | 89.83 | 9.14 | 1.03 |
| *d* log(*GDP*) | 14.49 | 81.89 | 3.62 | 15.66 | 83.06 | 1.28 | 16.59 | 83.14 | 0.27 |
| *d* log(*EXP*) | 7.71 | 39.37 | 52.92 | 18.41 | 10.11 | 71.48 | 16.62 | 5.44 | 77.94 |

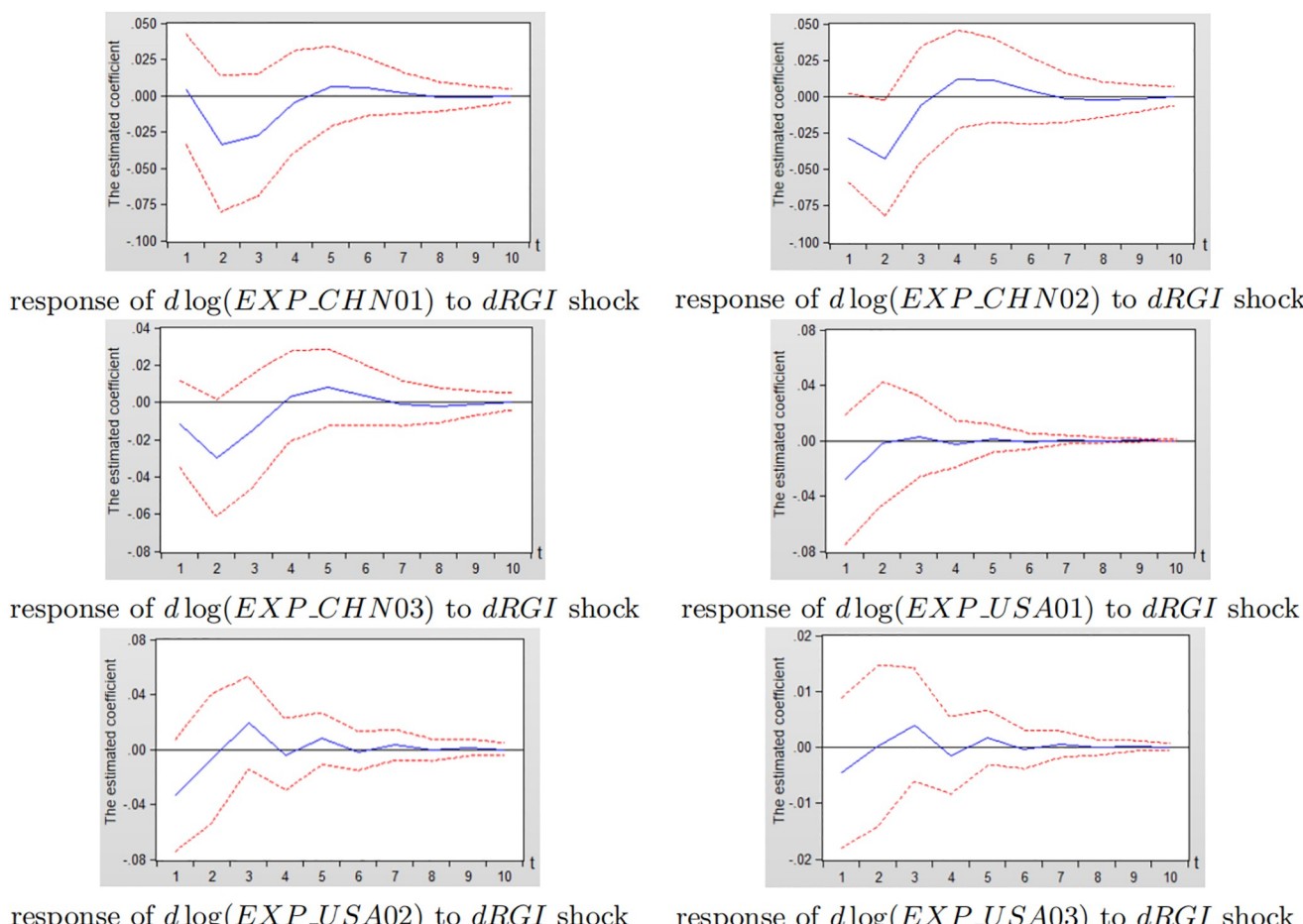

**Fig 9. Impulse response diagrams of reverse globalization to the exports of three major industries between China and the United States.**

force of US service industry exports, and the United States has extremely strong export competitiveness in technology-intensive service exports. Therefore, in the development of reverse globalization, the United States service industry exports are more sensitive to the impact of reverse globalization.

In summary, the inhibitory effect of reverse globalization on various subdivided industrial exports in China is significantly greater than that of the United States, which is consistent with the findings bewteen developing and developed countries in section 3.2. The inhibitory effect of reverse globalization on China's sub-manufacturing exports is slightly greater than sub-service exports, while the negative effects of reverse globalization on the United States sub-manufacturing exports and sub-service exports are not significantly different. Therefore, when China is responding to the impact of reverse globalization, in addition to maintaining the relative advantages in high-product-complexity manufacturing exports such as fabricated metal products, electrical and optical equipments, and machinery equipments, China still needs to improve the low-product-complexity manufacturing exports of non-metallic mineral products, textiles, food, and tobacco. Furthermore, China needs to actively increase the exports of water transportation, construction, land transportation and other related sub-service industry that are relatively less restrained by reverse globalization. While improving the sub-service exports such as construction, transportation and warehousing, information and

**Table 13. The classification of subdivided manufacturing and service industry.**

| Industry Number | Description | Code |
|---|---|---|
| 12 sub-manufacturing industry | | |
| 01 | Manufacture of food products, beverages and tobacco products | C10 − C12 |
| 02 | Manufacture of textiles, wearing apparel and leather products | C13 − C15 |
| 03 | Manufacture of wood and of products of wood and cork | C16 |
| 04 | Manufacture of paper, printing and publishing products | C17 − C18 |
| 05 | Manufacture of coke and refined petroleum products | C19 |
| 06 | Manufacture of chemicals and basic pharmaceutical products | C20 − C21 |
| 07 | Manufacture of rubber and plastic products | C22 |
| 08 | Manufacture of other non-metallic mineral products | C23 |
| 09 | Manufacture of basic metals and fabricated metal products | C24 − C25 |
| 10 | Manufacture of computer, electronic and optical products | C26 |
| 11 | Manufacture of electrical equipment, machinery and equipment | C27 − C28 |
| 12 | Manufacture of motor vehicles, trailers and other transport equipment | C29 − C30 |
| 14 sub-service industry | | |
| 01 | Electricity, gas and water supply, sewerage and waste collection | D35 + E36 − E39 |
| 02 | Construction | F |
| 03 | Wholesale trade and retail trade | G45 − G47 |
| 04 | Accommodation and food service activities | I |
| 05 | Land transportation and transportation via pipelines | H49 |
| 06 | Water transportation | H50 |
| 07 | Air transportation | H51 |
| 08 | Warehousing, postal and courier activities | H52 − H53 |
| 09 | Publishing, telecommunications and information service activities | J58 − J63 |
| 10 | Financial service and insurance activities | K64 − K66 |
| 11 | Real estate activities | L68 |
| 12 | Public administration and social security | O84 |
| 13 | Education | P85 |
| 14 | Human health and social work activities | Q |

telecommunications, the United States needs to actively promote the exports of wood, textiles, non-metallic minerals and other related sub-manufacturing products that are relatively less restrained by reverse globalization.

## 4. Discussion

This paper empirically analyzes the dynamic impact of reverse globalization on export trade from the perspectives of world, country, industry, subdivided manufacturing and service industry. The study finds that reverse globalization has a significant inhibitory effect on export trade, which is similar to the results of Garg and Sushil [5], Dür et al. [8], He et al. [10], etc. The innovation of this paper is mainly from the perspective of country level and compares the differential impact of reverse globalization on the exports of different country categories. This paper holds that reverse globalization hinders the development of export trade in EU countries, which partially supports the research results of Dür et al. [8] and Li et al. [12]. In addition, this paper concludes for the first time that developing and non-EU countries' exports are more affected by reverse globalization than those of developed countries and EU countries.

This paper finds that reverse globalization has a significant negative impact on a country's manufacturing exports, which confirms the research results of Garg and Sushil [5], Crino and

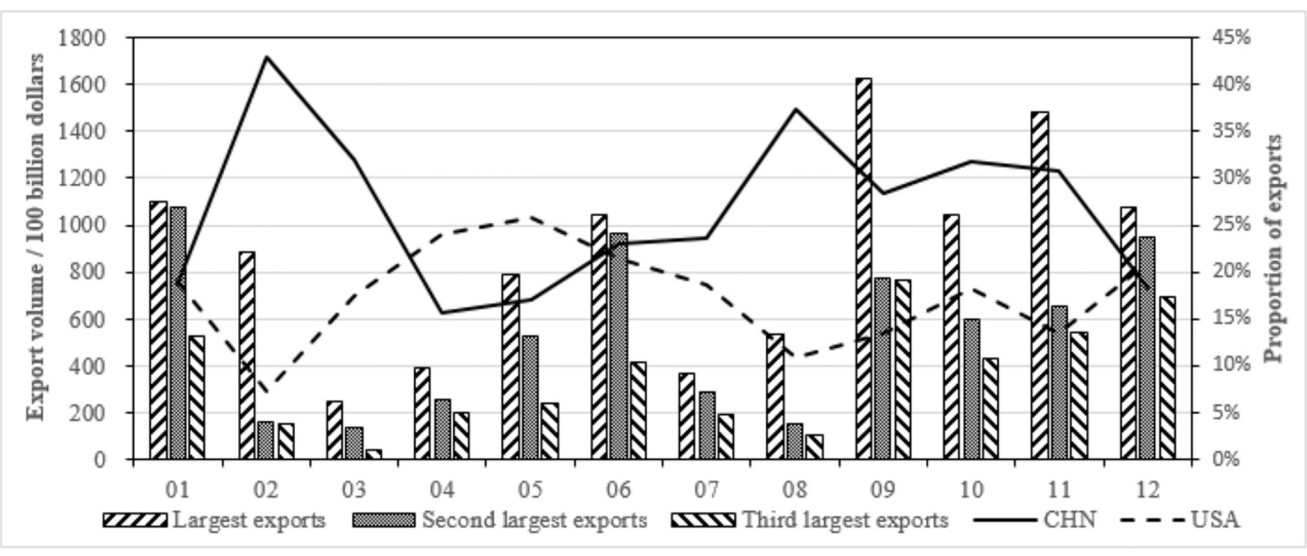

**Fig 10. Top three exports of subdivided manufacturing industry and the proportions between China and the United States.**

Epifani [9], He et al. [10], etc. Furthermore, the innovation of this paper is mainly from the perspective of industrial and subdivided industry to analyze the heterogeneous effects of reverse globalization on exports of different industry, subdivided manufacturing and service industry. The study finds that reverse globalization has the greatest inhibitory effect on the secondary industry exports, followed by the tertiary industry, and the inhibitory effect of reverse globalization on China's subdivided industry exports is significantly greater than that of the United States. Therefore, developing countries need to actively adjust their domestic industrial structure and export trade structure, enhance the export scale of industries with high technological complexity, and further increase trade cooperation with other countries.

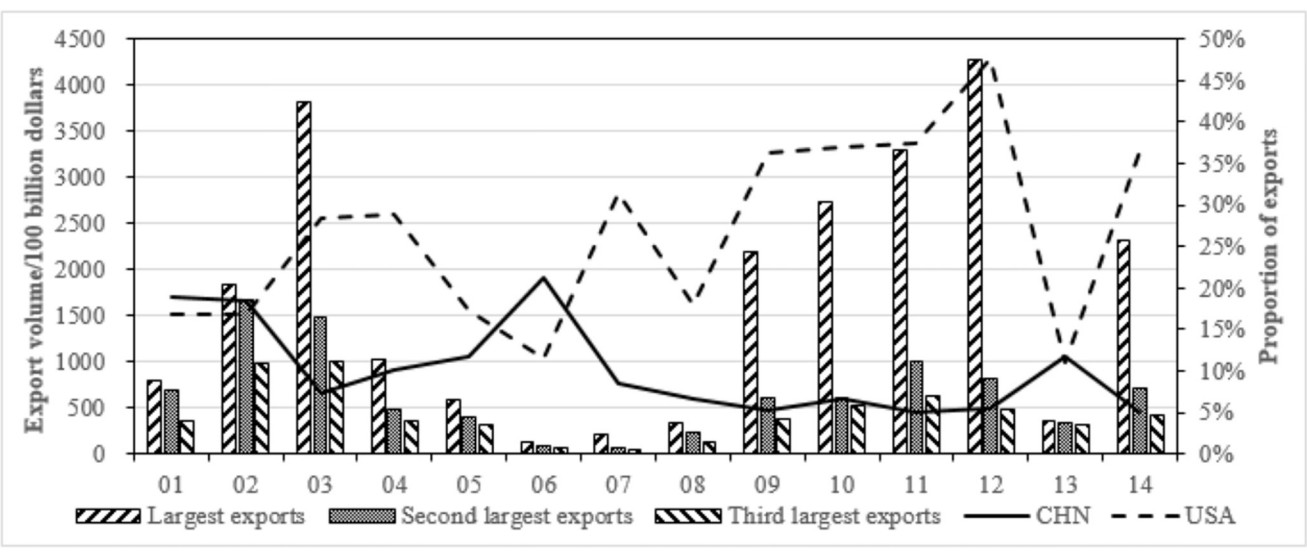

**Fig 11. Top three exports of subdivided service industry and the proportions between China and the United States.**

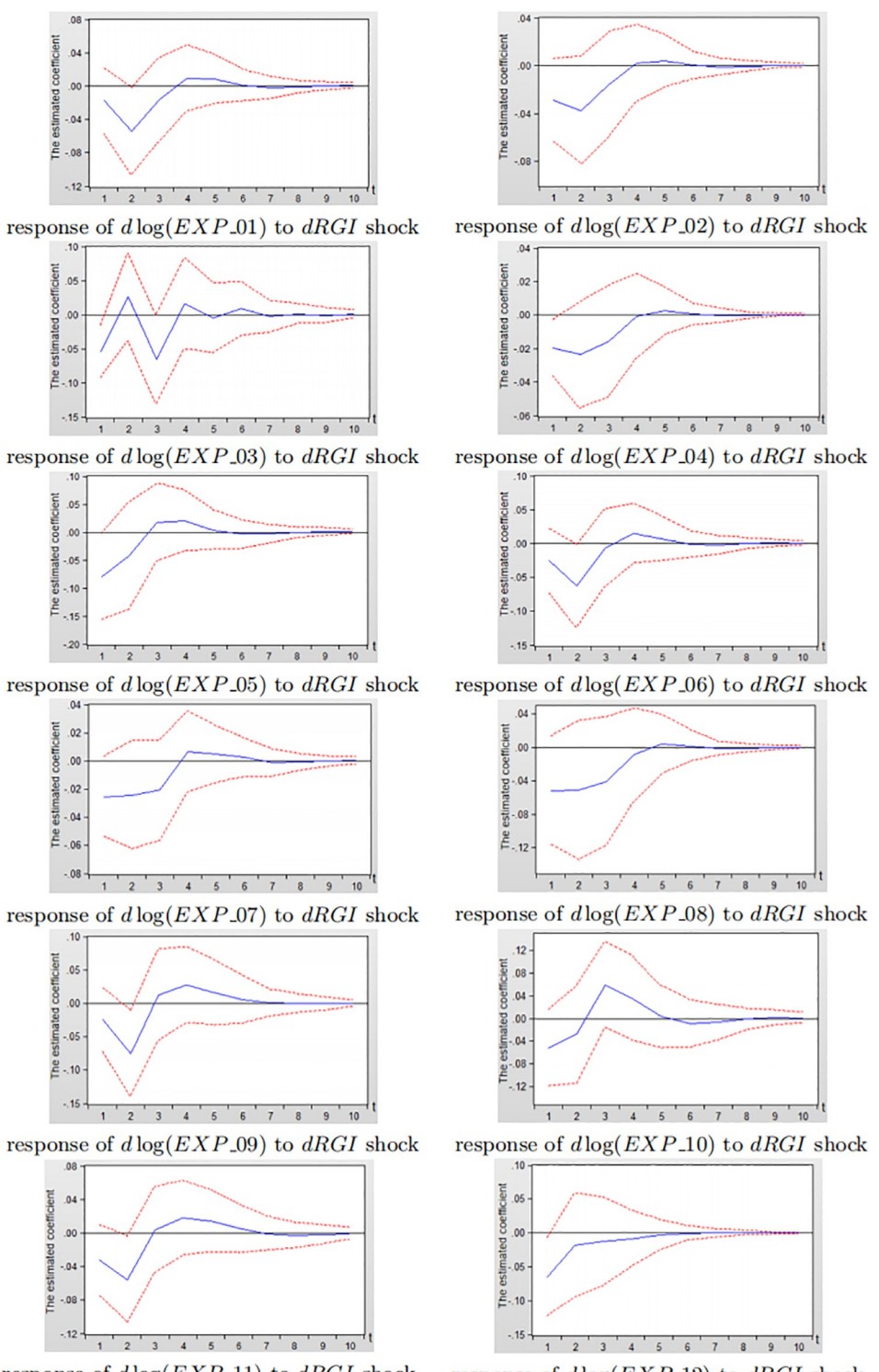

**Fig 12. Impulse response diagram of reverse globalization to China's subdivided manufacturing exports.**

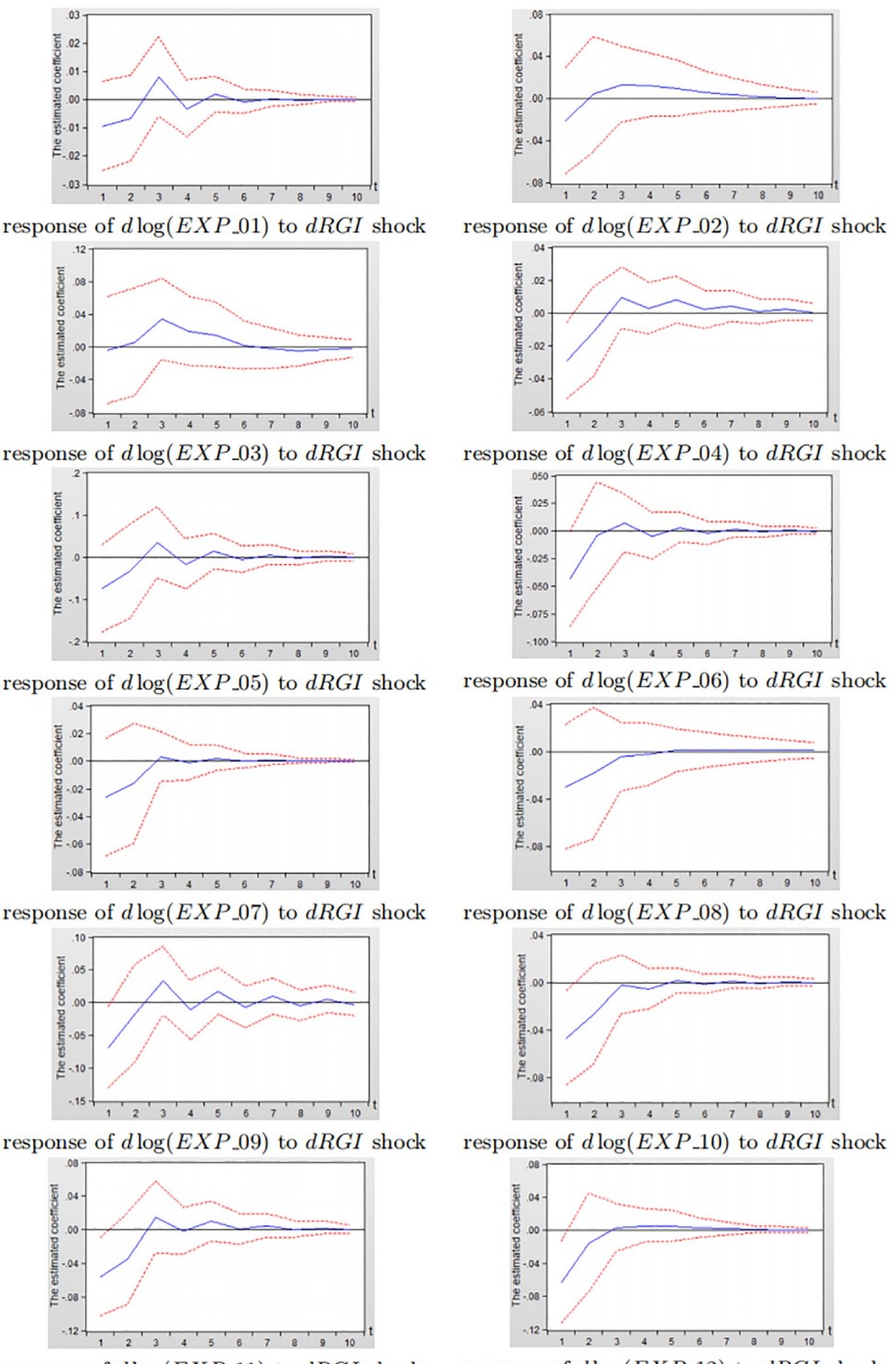

**Fig 13. Impulse response diagram of reverse globalization to the United States subdivided manufacturing exports.**

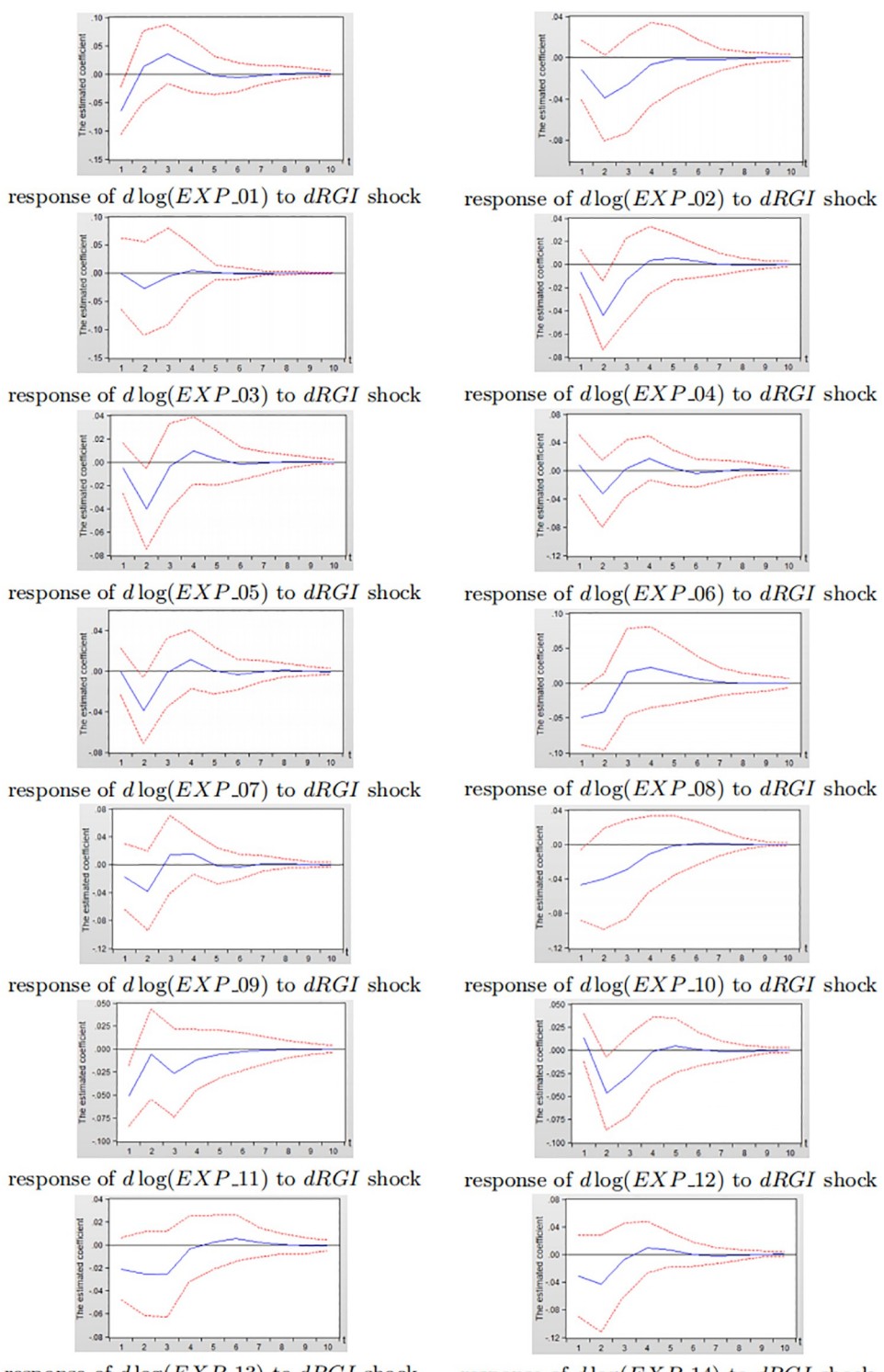

**Fig 14. Impulse response diagram of reverse globalization to China's sub-service exports.**

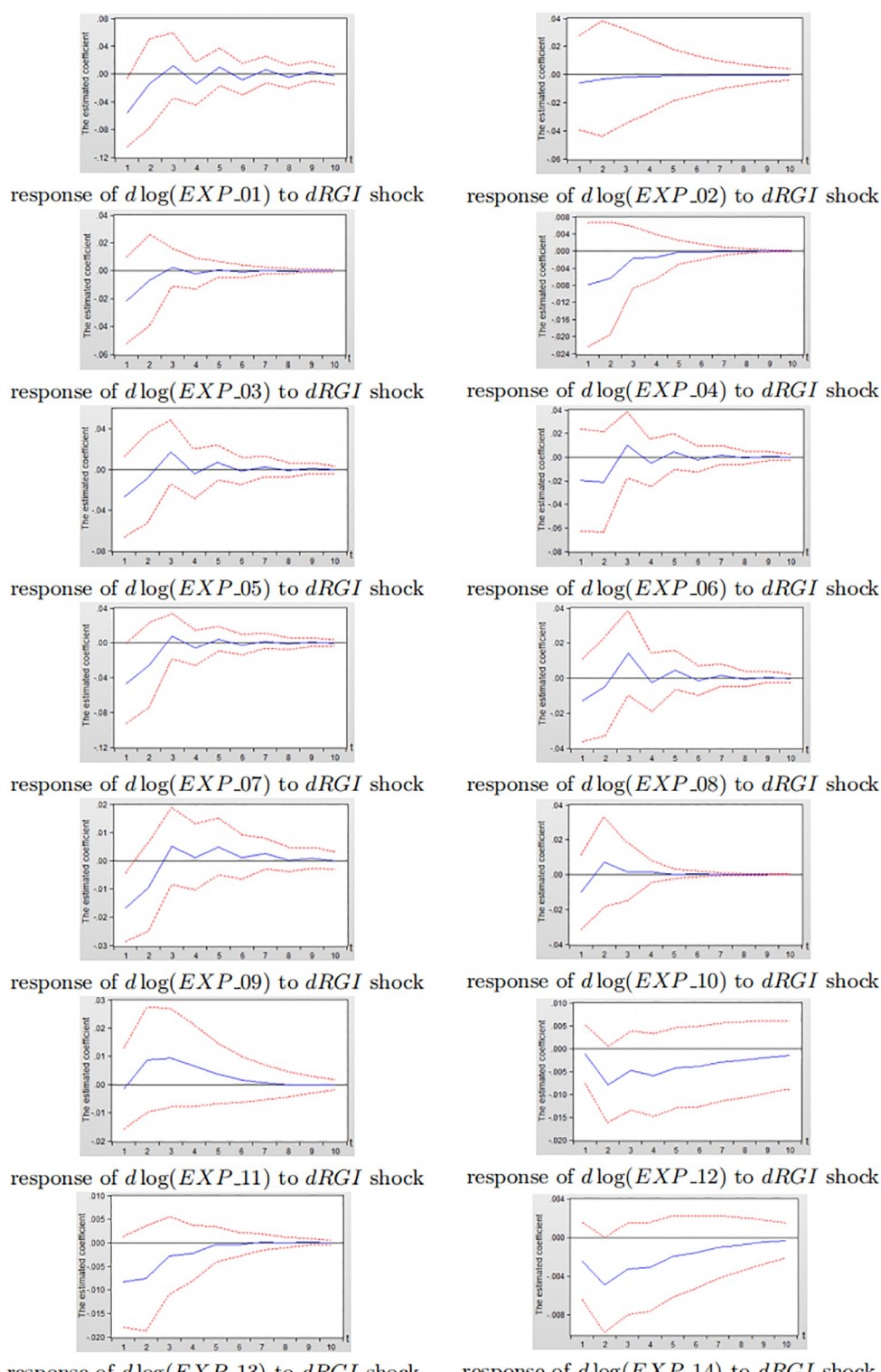

**Fig 15. Impulse response diagram of reverse globalization to the United States subdivided service exports.**

## 5. Conclusions and countermeasures

### 5.1 Conclusions

This paper uses the world input-output table in WIOD and the KOF globalization index to construct the time series and Panel VAR model from the perspectives of world, country, industry, subdivided manufacturing and service industry, and applies pvar2 estimation and Monte-Carlo simulation under GMM estimation to verify the dynamic impact of reverse globalization on a country's export trade transformation.

The main conclusions of this paper are as follows: Firstly, there is a significant non-linear negative effect of reverse globalization on economic scale and export trade. Compared with developed and EU countries, the exports of developing and non-EU countries are more affected by reverse globalization shocks. Secondly, reverse globalization has the greatest inhibition on the secondary industry exports, followed by the tertiary industry. And its suppressive effects on China's subdivided industry exports are significantly greater than that of the United States, but most of subdivided industry exports in the United States are more sensitive to the impact of reverse globalization. Thirdly, China's exports of high product complexity industry such as metal products, medicinal chemicals, electrical and optical products and mechanical equipments and other manufacturing industry are greatly affected by reverse globalization, and the exports of water transportation, construction, land transportation and other related service industry are relatively less restrained by reverse globalization.

### 5.2 Countermeasures

First, all countries should firmly support the concept of economic globalization and free trade, and jointly build an open world economy.

In facing the tide of reverse globalization, countries can achieve common development through international foreign direct investment and international trade cooperation, support the multilateral trading system, oppose any protectionist behavior, maintain global trade liberalization, actively promote the free flows of resources, technology and other production factors between regions, and improve the efficiency of resource allocation. It calls for strengthening the relevant regulations and functions of the World Trade Organization (WTO) to resolve international trade disputes in a fair and equal manner.

The United States, Europe and other countries need to strengthen international cooperation with trading partner with a more open attitude, and guide the healthy development of economic globalization.

Emerging market countries and developing countries should increase their representation and voice in global economic governance, and ensure that all countries have equal rights and opportunities in international economic cooperation.

Second, countries should actively carry out new types of trade cooperation, and propose national plans that are more conducive to global economic development.

All countries should improve the market allocation efficiency of factors such as labor, capital and technology in order to alleviate the inhibitory impact of reverse globalization on export trade.

For developing countries with advantages in population and resource endowments, market access conditions can be further relaxed to create a more attractive trade and investment environment. For example, as a country with large population and resources, China can rely on the Belt and Road Initiative and the New International Land-Sea Trade Corridor to establish a new trade partnership that is more equal and balanced, and enhance the scale of China's export trade to East Asia, Central Asia and Europe.

And for developing countries with geographical advantages, it is necessary to strengthen multilateral regional trade cooperation with neighboring countries in a more active and proactive manner, reduce the uncertainty of developing countries' participation in international trade, so as to buffer the inhibiting effects of reverse globalization on developing countries' export trade.

Third, countries should actively adjust the coordinated development of domestic industrial structure and export trade structure.

In the development of inverse globalization, countries should make full use of their comparative advantage industries, increase the investment of science and technology in the industrial structure, and continuously transform and adjust the export trade structure.

For developing countries with relatively low development of manufacturing and service industry, they should make full use of their relative advantageous resources to expand the scale of primary industry exports.

For developing countries with comparative advantages in manufacturing, in addition to maintaining the export advantage of industries with low-product-complexity, it is also necessary to improve the export scale of industries with high-technical-complexity, such as machinery manufacturing and transportation equipment. For example, China can improve the international competitiveness of industrial exports through technological upgrading and industrial restructuring. If the current industry exports are affected by reverse globalization and cannot participate in international trade cooperation, we can make full use of domestic market advantages to lead domestic consumption or realize industrial transformation, produce goods more needed in the domestic market, and realize the new development pattern of dual cycles of domestic and international economy.

In addition, for developing countries with absolute advantage in manufacturing, on the basis of stabilizing the development of light industry, they should improve the export of service products such as construction, transportation, finance and communication, and promote the adjustment and upgrading of domestic industrial structure, so as to alleviate the inhibiting effects of reverse globalization on the export of advantageous industries.

## Supporting information

**S1 Appendix. List of 42 countries and one region.**
(PDF)

**S1 Table. Goodness of fit test results under different variable forms.**
(PDF)

## Author Contributions

**Conceptualization:** Xueyan Wang, Weidong Meng, Bo Huang.

**Data curation:** Xueyan Wang.

**Formal analysis:** Xueyan Wang, Weidong Meng, Bo Huang.

**Funding acquisition:** Weidong Meng, Bo Huang, Yuyu Li.

**Methodology:** Xueyan Wang, Chunyang Wang.

**Software:** Xueyan Wang.

**Writing – original draft:** Xueyan Wang, Chunyang Wang, Yuyu Li.

**Writing – review & editing:** Xueyan Wang, Weidong Meng, Chunyang Wang, Bo Huang.

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
