## [Decision Letter · Decision Letter 0]

6 Mar 2022

PONE-D-21-35675Export trade structure transformation and countermeasures in the context of reverse globalizationPLOS ONE

Dear Dr. Huang,

Thank you for submitting your manuscript to PLOS ONE. After careful consideration, we feel that it has merit but does not fully meet PLOS ONE’s publication criteria as it currently stands. Therefore, we invite you to submit a revised version of the manuscript that addresses the points raised during the review process.

We look forward to receiving your revised manuscript.

Kind regards,

Ming Zhang, Ph.D.

Academic Editor

PLOS ONE

Journal Requirements:

"This study are supported by the Cultural Experts Project (The Propaganda Department of the Central Committee of the CPC [2016]133), the Natural Science Foundation of Chongqing (No.cstc2019jcyj-msxmX0616) and the Fundamental Research Funds for the Central Universities (No.2020CDJSK02PT12). The funders had no role in study design, data collection and analysis, decision to publish, or preparation of the manuscript."

"We are also grateful for the financial support of the Cultural Experts Project (The 543

Propaganda Department of the Central Committee of the CPC [2016]133), the Natural 544

Science Foundation of Chongqing (No. cstc2019jcyj-msxmX0616) and the Fundamental 545

Research Funds for the Central Universities (No. 2020CDJSK02PT12)"

Reviewers' comments:

Reviewer's Responses to Questions

**Comments to the Author**

1. Is the manuscript technically sound, and do the data support the conclusions?

Reviewer #1: Yes

Reviewer #2: Yes

2. Has the statistical analysis been performed appropriately and rigorously? 

Reviewer #1: Yes

Reviewer #2: Yes

3. Have the authors made all data underlying the findings in their manuscript fully available?

Reviewer #1: Yes

Reviewer #2: Yes

4. Is the manuscript presented in an intelligible fashion and written in standard English?

Reviewer #1: Yes

Reviewer #2: Yes

5. Review Comments to the Author

Reviewer #1: This paper uses the KOF Globalization Index and the world input-output tables in World Input-Output Database (WIOD), and empirically studies the transformation of a country's export trade and export structure in the context of reverse globalization from the perspectives of world, country, industry, subdivided manufacturing and service industry. The research method of this paper is basically appropriate, and the research content has certain practical significance. However, the description of the paper is slightly redundant, and the content of chapters is repeated and mixed. The logic of the thesis needs to be further sorted out, and the innovation needs to be further strengthened. It is recommended that the paper should be re-reviewed after major revisions. The specific comments are as follows:

(1) Line6, Line16, line17, line20, line28, it is suggested that the reference number should not be used as the subject, and there are many similar problems in the following;

(2) (2) Line14-51, the logic level of this introduction is not very clear. References on the causes, manifestations, countermeasures, and impacts of deglobalization are mixed, and the order is not very clear. Most of the references are simply listed, lacking conciseness, summary, and discussion. And the natural segment is too long and should be segmented.

(3) Line66-69, not necessary, could be deleted.

(4) Part 2. The definition and current status of reverse globalization. This part is more like a literature review, which has a lot of repetition with the introduction and lacks a good concise summary and discussion. Line103-106 introduces the reverse globalization measurement index proposed in this paper. Line106-112 lists the references, first introduce the globalization index proposed in the references, then describes the impact of economic globalization, and finally introduces the index proposed in the literature again. Line112 then explains the index proposed in this paper. The logic is a little confused. How many indicators described in this part are related to Fig1? The description is a little confusing. Generally speaking, there are a lot of process descriptions in this part of the method, but the result description is too simple, which does not well describe the current situation of reverse globalization, and the overall description logic is not very clear.

(5) Fig1 needs to increase the resolution to increase readability.

(6) Line167, which does not give the specific name or location diagram of the selected country.

(7) Line167, WIOD is the data in 2016, and the time to 2018 (line197) is discussed later, please explain.

(8) Line169, Taiwan cannot be connected after countries. Taiwan is a part of China, not a country. The writing of this article is ambiguous and must be cautious.

(9) Table 1, the time length range of data should be described.

(10) Line199-201, please present the test results.

(11) Fig2, fig10, fig13-16, complete drawing elements need to be supplemented.

(12) Line238, should it be a developing country?

(13) Fig3-9, fig11-12, please increase the picture resolution to increase readability.

(14) Part 4. Empirical result. The analysis methods used in this part are completely consistent from the international, national, industrial, subdivided manufacturing, and service industries. The description of each result is too detailed, which makes this chapter repetitive and redundant. The authors need to refine and summarize the research results better.

(15) Part5.2 Countermeasures. The second and third sections are specific to the measures China can take. Relevant recommendations for other developing countries should be added to echo the previous research perspectives based on developing countries and to make the recommendations more extensive and universal.

Reviewer #2: 1. This paper lacks theoretical analysis. Before establishing VAR model, it is necessary to theoretically explain the relationship among RGI, GDP and EXP. In addition, the author needs to explain why only these three variables are selected and other variables are not selected.

2. This paper only makes unit root test for variables, and lacks cointegration test.

3. This paper only lists the empirical results and lacks in-depth analysis of the empirical results.

4. This paper is lack of discussion section. It is suggested that the author compare the similarities and differences between the research results of this paper and the existing research results.

6. PLOS authors have the option to publish the peer review history of their article (what does this mean?). If published, this will include your full peer review and any attached files.

Reviewer #1: **Yes: **Binggeng Xie

Reviewer #2: No

---

## [Author Response · Author response to Decision Letter 0]

21 Apr 2022

Response to Editors’ and Reviewers’ Comments

Title: Export trade structure transformation and countermeasures in the context of reverse globalization

Number: PONE-D-21-35675

 Thanks for Editors’ and Reviewers’ valuable suggestions. The authors carefully improved the paper with red font according to the suggestions as follows:

Response to journal requirements:

Suggestion 1:

Please ensure that your manuscript meets PLOS ONE’s style requirements, including those for file naming. The PLOS ONE style templates can be found at https://journals.plos.org/plosone/s/file?id=wjVg/PLOSOne_formatting_sample_main_body.pdf and https://journals.plos.org/plosone/s/file?id=ba62/PLOSOne_formatting_sample_title_authors_affiliations.pdf.

Response:

　　We checked original manuscript, and made sure that our manuscript meets PLOS ONE’s style requirements, including those for file naming.

Suggestion 2:

Thank you for stating the following in the Acknowledgments Section of your manuscript:

“This study are supported by the Cultural Experts Project (The Propaganda Department of the Central Committee of the CPC [2016]133), the Natural Science Foundation of Chongqing (No.cstc2019jcyj-msxmX0616) and the Fundamental Research Funds for the Central Universities (No.2020CDJSK02PT12). The funders had no role in study design, data collection and analysis, decision to publish, or preparation of the manuscript.”

“We are also grateful for the financial support of the Cultural Experts Project (The 543 Propaganda Department of the Central Committee of the CPC [2016]133), the Natural 544 Science Foundation of Chongqing (No. cstc2019jcyj-msxmX0616) and the Fundamental 545 Research Funds for the Central Universities (No. 2020CDJSK02PT12).”

Response:

 We removed the funding-related text from our manuscript and included the amended Funding Statement within our cover letter.

Response to the comments of reviewer #1:

Suggestion 1:

 Line6, Line16, line17, line20, line28, it is suggested that the reference number should not be used as the subject, and there are many similar problems in the following.

Response:

 We checked and modified the citation format of full-text references, including line6, line16, line17, Line20, line28, etc. The detailed modifications are as follows:

 Changed “[1]” into “Postelnicu et al. [1]”, changed “[3]” into “Chen and Hsu [3]”, changed “[4]” into “James [11]”, changed “[5]” into “Li et al. [12]”, changed “[10]” into “Zhou et al. [6]”, changed “[11]” into “Crino and Epifani [9]”, changed “[17]” into “Samimi and Jenatabadi [24]”, changed “[18]” into “Xu et al. [7]”, changed “[20]” into “Dreher [22]”, changed “[21]” into

“Postelnicu et al. [25]”, changed “[22]” into “Gygli et al. [26]”, changed “[23]” into “Dreher et al. [27]”, changed “[24]” into “Sims [13]”, changed “[25]” into “Anderson and Hsiao [14]”, changed “[26]” into “Holtz-Eakin et al. [15]”, changed “[30]” and “[32]” into “Love and Zicchino [31]” and “Shank and Vianna [33]”, and changed “[44]” into “Wang et al. [45]”.

 Please see the whole manuscript.

Suggestion 2:

 Line14-51, the logic level of this introduction is not very clear. References on the causes, manifestations, countermeasures, and impacts of deglobalization are mixed, and the order is not very clear. Most of the references are simply listed, lacking conciseness, summary, and discussion. And the natural segment is too long and should be segmented.

Response:

 We sorted out the logical structure of the references in the introduction section and rearranged the order of the literature. Firstly, the causes and manifestations of reverse globalization were deleted, which were less relevant to the theme of this paper, and the focus was on introducing the impact of reverse globalization on the export trade structure and countermeasures.

Secondly, the literature review in the introduction section was divided into two paragraphs. The first paragraph introduced the relevant researchs on the impact of reverse globalization on trade, the second one introduced the countermeasure studies on reverse globalization, and summarized and discussed the references. Finally, based on the review of the existing literature, the description of the innovation was further improved by comparing and summarizing the inadequacies of the existing literatures. The detailed modifications are as follows:

 ... ...

 Reverse globalization is not simply deglobalization, but mainly the phenomenon of market protection in different degrees and forms after the development of economic globalization to a certain stage, which may lead to greater competition among big countries [3,4]. Scholars have studied the impact of reverse globalization on economic growth, investment and trade. Among them, Garg and Sushil [5] analyzed the determinants of reverse globalization, such as global economy, income inequality, technological development, and other factors. Zhou et al. [6] studied the short-term and long-term impact of external reverse globalization on China’s macroeconomic performance and economic growth. Xu et al. [7] found a significant negative effect of de-globalization behavior on international investment. In the development of reverse globalization, developed countries put forward more stringent trade barriers, which led to the contradiction of EU in pursuing trade interests and its bargaining power in trade negotiations, thereby restricting EU from changing the trade agreements [8], and the uncertainty of international trade environment affected the dynamic changes of a country’s trade structure [9]. Using China’s manufacturing exports as an example, He et al. [10] found that de-globalization leads to an increase in manufacturing export trade costs, which did not affect China’s coordinated regional development, but is not conducive to manufacturing upgrading and China’s economic

transformation. Although the above literature has studied the impact of reverse globalization on economic growth and trade, it does not analyze the heterogeneous impact of reverse globalization on export trade in different countries, industries and subsectors.

 In addition, some scholars have studied the countermeasures to deal with the impact of reverse globalization. James [11] considered that the decrease in cross-border capital flows and the slowdown in world trade growth accelerated the development of reverse globalization, while the effective reduction of trade barriers related to geographical distance and artificially high barriers can better cope with the uncertainty of international trade cooperation caused by reverse globalization [12]. He et al. [10] believed that China should improve regional transportation and communication infrastructure, reduce geographical trade costs, facilitate enterprises access to export market, maintain the export scale of existing exporters, and improve the competitiveness of enterprises to explore foreign markets, so as to reduce the impact of reverse globalization on China’s manufacturing export trade. In response to the impact of reverse globalization, Garg and Sushil [5] emphasized that governments should focus on domestic manufacturing and production, strengthen policy support for local manufacturing industry, and further expand the share of sales in local markets, so as to reduce technological dependence on other countries or regions. Although the above-mentioned literature studied the countermeasures to reverse globalization, it does not give the relevant countermeasures of reverse globalization on the transformation of export trade structure of each subdivided manufacturing industry and service industry.

 This paper makes three main contributions. First, most of the existing literatures discuss reverse globalization from the qualitative level, while we analyze the impact and countermeasures of reverse globalization on export trade structure from the empirical level. From the perspectives of world, country, industry, subdivided manufacturing and service industry, this paper quantitatively analyzes the dynamic impact of reverse globalization on export trade by using time series and panel data structure, and proposes some suggestions to deal with the impact of reverse globalization. Second, we obtain valuable research results through empirical analysis. The study finds that compared to developed countries and EU countries, the exports of developing countries and non-EU countries are more affected by reverse globalization. Reverse globalization has the greatest inhibitory effect on the secondary industry exports, and the restraint effect on China’s subdivided manufacturing and service industry exports is significantly greater than that of the United States. Third, through the research results, we give relevant suggestions to deal with reverse globalization. All countries need to strengthen international trade cooperation with trading partners with a more open attitude, and improve the spatial allocation efficiency of resources, technology, talents and other elements. Developing countries can take the lead in international trade cooperation on a small scale based on their geographical location, resource endowment and other advantages, and actively adjust the coordinated development of the industrial structure and export trade structure of various country, so as to alleviate the inhibitory effect of reverse globalization on export trade.

 The added references as follows:

1) Garg S, Sushil. Determinants of deglobalization: A hierarchical model to explore their interrelations as a conduit to policy. Journal of Policy Modeling. 2021;43(2):433–447.

2) He LY, Lin X, Zhang ZX. The impact of de-globalization on China’s economic transformation: Evidence from manufacturing export. Journal of Policy Modeling. 2020;42(3):628–660.

 Please see the sections of “Introduction” and “References”.

Suggestion 3:

 Line66-69, not necessary, could be deleted.

Response:

 We deleted line66-69 from our manuscript.

 Please see section 1.

Suggestion 4:

 Part 2. The definition and current status of reverse globalization. This part is more like a literature review, which has a lot of repetition with the introduction and lacks a good concise summary and discussion. Line103-106 introduces the reverse globalization measurement index proposed in this paper. Line106-112 lists the references, first introduce the globalization index

proposed in the references, then describes the impact of economic globalization, and finally introduces the index proposed in the literature again. Line112 then explains the index proposed in this paper. The logic is a little confused. How many indicators described in this part are related to Fig1? The description is a little confusing. Generally speaking, there are a lot of process descriptions in this part of the method, but the result description is too simple, which does not well describe the current situation of reverse globalization, and the overall description logic is not very clear.

Response:

 Combined with Suggestion 2, we deleted the part of “2. The definition and current status of reverse globalization” and organized the relevant literature into “Introduction” with a concise summary and discussion. The measurement of reverse globalization index proposed in this section was placed in “Model construction and variable selection”, and the serial numbers of

each chapter were modified accordingly. For example:

 Changed “3. Model, variable and methodology” into “2. Model, variable and methodology”, changed “3.1 Model construction and variable selection” into “2.1. Model construction and variable selection”, changed “3.2 Methodology selection” into “2.2. Methodology selection”, changed “4. Empirical results” into “3. Empirical results”, etc.

 The specific modifications are as follows:

 ... ...

 The existing literatures failed to directly quantify the indicator of reverse globalization. In order to better evaluate the dynamic impact of reverse globalization on export trade structure, this paper uses globalization indicator for reference to measure reverse globalization. Dreher [22] firstly proposed the globalization index covering economic, social and political aspects, and Miskiewicz and Ausloos [23] defined four macroeconomic indexes to describe the process of globalization. Samimi and Jenatabadi [24], Potrafke [25] both found that economic globalization promoted economic growth, but had a differential impact on income. Gygli et al. [26] introduced the revised version of the KOF globalization index, Dreher et al. [27] updated the degree of globalization of 195 countries or regions in the world at the economic, social and political levels since 1970, and the KOF globalization index had become the most widely used globalization indicator in academia. Therefore, this paper uses the KOF globalization index as the basic data, excluding the trade globalization index in the sub-dimension of economic globalization, and takes the inverse of the globalization index to obtain the Reverse Globalization Index (RGI), which is used to measure the performance of reverse globalization at the level of institutional factors such as tariffs, finance, interpersonal relationships, information, culture and politics.

 ... ...

 The added references as follows:

Miskiewicz J, Ausloos M. Has the world economy reached its globalization limit? Physica A: Statistical Mechanics and its Applications. 2010;389(4):797–806.

 Please see the sections of “Introduction”, “Model construction and variable selection” and “References”.

Suggestion 5:

 Fig1 needs to increase the resolution to increase readability.

Response:

 Combined with Suggestion 4, we deleted Fig 1 and the conclusion of Fig 1 was placed in “Introduction”. The specific modifications are as follows:

 ... ... The frequent trade wars and other issues show the decreasing economic interdependence among countries and the significant trend of reverse globalization. ... ...

 Please see section 1.

Suggestion 6:

 Line167, which does not give the specific name or location diagram of the selected country.

Response:

 We changed “43 countries (28 EU countries and 15 other major countries)” in Line167 into “42 countries and one region (See S1 Appendix for the specific name or location)”.

 S1 Appendix of Supporting Information Section is in line585.

 Please see section 2.1 and “Supporting Information”.

Suggestion 7:

 Line167, WIOD is the data in 2016, and the time to 2018 (line197) is discussed later, please explain.

Response:

 We explained the reasons for choosing the two time dimensions of 2016 and 2018 in line140-146 and line170-171. The detailed supplements are as follows:

 ... ... Since the database is only updated to the WIOD November 2016 Release, this paper mainly uses the industrial export data in world input-output tables published by WIOD in 2016 (see, [28,29]) to study the impact of reverse globalization on export trade and export structure of industries, subdivided manufacturing and service industries for the period from 2000 to 2014. The table covers 42 countries and one region (See S1 Appendix for the specific name or location), and each country or region consists of 56 components. ... ...

 ... ...

 To study the dynamic impact of reverse globalization on export trade structure transformation, we first consider the analysis at the world level. The world-level export data for this part comes from the World Bank (2000-2018). ... ...

 Please see section 2.1 and 3.1.

Suggestion 8:

 Line169, Taiwan cannot be connected after countries. Taiwan is a part of China, not a country. The writing of this article is ambiguous and must be cautious.

Response:

 We deleted “(except Taiwan)” in the original manuscript, and classified Taiwan as a region in the main text. The specific modifications are as follows:

 ... ... The table covers 42 countries and one region (See S1 Appendix for the specific name or location), and each country or region consists of 56 components. ... ...

 Please see section 2.1.

Suggestion 9:

 Table 1, the time length range of data should be described.

Response:

 We supplemented Table 1 with the time length range of data for each variable. Among them, the time length range of RGI is 2000-2018, GDP is 2000-2018, and EXP is 2000-2018 (Word bank) and 2000-2014 (WIOD). The detailed modifications are as follows:

 Table 1. Variable interpretation and data source. shows in Page5.

 Please see section 2.1.

Suggestion 10:

 Line199-201, please present the test results.

Response:

 We presented the test results of line199-201 (See S1 Table), and modified the result analysis of Table 2 as follows:

 From the Skewness and Kurtosis statistics, the conditional distributions of GDP and EXP are skewed. After log(GDP) and log(EXP) processing, the nature and correlation of the data will not be changed, but it narrows the range of variables and makes the data more stable. Although the variables log(GDP) and log(EXP) are still biased, the accompanying probability values of the two are 0.3682 and 0.3041, which are both greater than the significance level of 0.05 or even 0.1, indicating that the variables log(GDP) and log(EXP) obey the normal distribution. In addition, taking the logarithm of variables GDP and EXP can improve the effect of parameter estimation and the goodness of fit of the model (See S1 Table). Therefore, the subsequent empirical analysis directly takes the logarithm of GDP and EXP .

 S1 Table of Supporting Information Section is in line588.

 Please see section 3.1 and “Supporting Information”.

Suggestion 11:

 Fig2, fig10, fig13-16, complete drawing elements need to be supplemented.

Response:

 We supplemented the drawing elements such as the names of horizontal and vertical coordinates in fig2, fig10 and fig13-16 of the original manuscript. According to Suggestion 4, we deleted Fig1, so we changed “Fig2, fig10, fig13-16” into “Fig1, Fig9, Fig12-15”. The detailed supplements are shown in the manuscript's Fig1, Fig9 and Fig12-15.

 Please see Fig1, Fig9, Fig12-15.

Suggestion 12:

 Line238, should it be a developing country?

Response:

 We changed “developed country” in line238 into “developing country”.

 Please see section 3.1.

Suggestion 13:

 Fig3-9, fig11-12, please increase the picture resolution to increase readability.

Response:

 The resolution of Fig3-9, Fig11-12 in original manuscript had been improved as required. According to Suggestion 4, we deleted Fig1, so we changed “Fig3-9, fig11-12” into “Fig2-8, Fig10-11”. The detailed modifications are shown in the manuscript's Fig2-8 and Fig10-11.

 Please see Fig2-8 and Fig10-11.

Suggestion 14:

 Part 4. Empirical result. The analysis methods used in this part are completely consistent from the international, national, industrial, subdivided manufacturing, and service industries. The description of each result is too detailed, which makes this chapter repetitive and redundant. The authors need to refine and summarize the research results better.

Response:

 We deleted the repetitive and redundant expressions in the original “Empirical result”, and refined and summarized the research results. The detailed modifications are as follows:

 (1) We changed “Table 5” in line248 into “Table 6”. Line250-256 is modified as follows:

 From the unit root test results of the variables under different country groupings in Table 6, it is clear that, the probability p values of 1st difference series of RGI, log(GDP) and log(EXP) in all four unit root tests are less than 0.10, indicating that the 1st difference series of the three variables are stationary. Therefore, the 1st difference variables of RGI, log(GDP) and log(EXP) are used in the construction of the Panel V AR model at the country level.

 (2) Result analysis for Fig 2 and Fig 3 as follows:

 Fig 2 and Fig 3 describe the impulse response diagram from developing and developed countries. Like the analysis at the world level, the impact of a country’s economic scale and export trade under different country groupings on reverse globalization is close to 0. In addition, a standard deviation shock of reverse globalization has a significant negative effect on a country’s economic scale and export, and has a greater effect on the country’s export trade. ... ...

 (3) Result analysis for Fig 4 and Fig 5 as follows:

 Fig 4 and Fig 5 describe the impulse response diagram of non-EU and EU countries, respectively. The figures show that a country’s economic scale and export trade have no effect on reverse globalization. Comparing the results of Cluster 3 and 4, it is found that the inhibitory effect of reverse globalization on the export of non-EU countries is higher than that of EU countries. ... ...

 (4) We changed “Table 9” in line330 into “Table 11”. Line327-330 is modified as follows:

 ... ... The results of pvar2 estimation under different industry groups are presented in Table 11.

 (5) We changed “Table 10” in line352 into “Table 12”. Line353-358 is modified as follows:

 ... ... The results find that the explanatory power of reverse globalization on exports of the three major industries is 7.71%, 18.41% and 16.62%, respectively, i.e., the impact of reverse globalization has a greater inhibitory effect on the secondary industry exports, which is consistent with the pvar2 estimation and impulse response results.

 Please see section 3.

Suggestion 15:

 Part5.2 Countermeasures. The second and third sections are specific to the measures China can take. Relevant recommendations for other developing countries should be added to echo the previous research perspectives based on developing countries and to make the recommendations more extensive and universal.

Response:

 We added relevant recommendations for other developing countries and made them more universal. The specific modifications are as follows:

 (1) In the second section：

 Second, countries should actively carry out new types of trade cooperation, and propose national plans that are more conducive to global economic development.

 All countries should improve the market allocation efficiency of factors such as labor, capital and technology in order to alleviate the inhibitory impact of reverse globalization on export trade.

 For developing countries with advantages in population and resource endowments, market access conditions can be further relaxed to create a more attractive trade and investment environment. For example, as a country with large population and resources, China can rely on the Belt and Road Initiative and the New International Land-Sea Trade Corridor to establish a new trade partnership that is more equal and balanced, and enhance the scale of China’s export trade to East Asia, Central Asia and Europe.

 And for developing countries with geographical advantages, it is necessary to strengthen multilateral regional trade cooperation with neighboring countries in a more active and proactive manner, reduce the uncertainty of developing countries’participation in international trade, so as to buffer the inhibiting effects of reverse globalization on developing countries’ export trade.

 (2)In the third section：

 Third, countries should actively adjust the coordinated development of domestic industrial structure and export trade structure.

 In the development of inverse globalization, countries should make full use of their comparative advantage industries, increase the investment of science and technology in the industrial structure, and continuously transform and adjust the export trade structure.

 For developing countries with relatively low development of manufacturing and service industry, they should make full use of their relative advantageous resources to expand the scale of primary industry exports.

 For developing countries with comparative advantages in manufacturing, in addition to maintaining the export advantage of industries with low-product-complexity, it is also necessary to improve the export scale of industries with high-technical-complexity, such as machinery manufacturing and transportation equipment. For example, China can improve the international competitiveness of industrial exports through technological upgrading and industrial restructuring. If the current industry exports are affected by reverse globalization and can not participate in international trade cooperation, we can make full use of domestic market advantages to lead domestic consumption or realize industrial transformation, produce goods more needed in the domestic market, and realize the new development pattern of dual cycles of domestic and international economy.

 In addition, for developing countries with absolute advantage in manufacturing, on the basis of stabilizing the development of light industry, they should improve the export of service products such as construction, transportation, finance and communication, and promote the adjustment and upgrading of domestic industrial structure, so as to alleviate the inhibiting

effects of reverse globalization on the export of advantageous industries.

 Please see section 5.2.

Response to the comments of reviewer #2:

Suggestion 1:

 This paper lacks theoretical analysis. Before establishing VAR model, it is necessary to theoretically explain the relationship among RGI, GDP and EXP. In addition, the author needs to explain why only these three variables are selected and other variables are not selected.

Response:

 We added a theoretical analysis in Section 2.1 to explain the relationship among RGI, GDP and EXP, and gave the reasons for selecting these three variables. The detailed additions are as follows:

 ... ...

 In the development of reverse globalization, developed countries often adopt stricter international trade barriers to protect their domestic markets [16], reduce import trade to developing countries and emerging countries, and encourage overseas multinational enterprises or factories to return home to enhance the market competitiveness of their goods [17,18]. In consideration of national interests and geopolitics, developed countries gradually squeeze the development space of emerging market countries, hindering the free flow of global capital, resources, technology and other production factors, which eventually leads to a gradual decrease in the share of international trade and foreign direct investment flows among countries [19-21]. Therefore, reverse globalization has an inhibitory effect on a country’s level of economic development and

export trade.

 ... ...

 Before applying the VAR model for empirical analysis, it is necessary to obtain the stationary series of each variable and the optimal lags of the model, so as to ensure the effectiveness of the model estimation results and there is no autocorrelation of the residuals. When there are K variables in the model and the optimal lag order is P, the final estimated parameters obtained by using the VAR model are (P ∗ K2 + K), which also includes the variance and covariance matrix of the residual [15]. Since the variables in VAR model should not be selected too much, in order to study the differential impact of a country’s export structure transformation under the background of reverse globalization, this paper selects a three-variable vector, including reverse globalization index, country’s economic scale, and export trade, denoted as [RGI, GDP, EXP]. ... ...

 The added references as follows:

1) Fredi-Sanchez JLM. Deglobalization and public diplomacy. International Journal of Communication. 2021;15:905–926.

2) Paul TV. Globalization, deglobalization and reglobalization: Adapting liberal international order. International Affairs. 2021;97(5):1599–1620.

3) Ripsman NM. Globalization, deglobalization and Great Power politics. International Affairs. 2021;97(5):1317–1333.

4) Butzbach O, Fuller DB, Schnyder G. Manufacturing discontent: National institutions, multinational firm strategies, and anti-globalization backlash in advanced economies. Global Strategy Journal. 2020;10(1):67–93.

5) Du WP, Yan HM, Feng ZM, Yang ZQ, Yang YZ. The external dependence of ecological products: Spatial-temporal features and future predictions. Journal of Environmental Management. 2022;304:114190.

 Please see section 2.1 and “References”.

Suggestion 2:

 This paper only makes unit root test for variables, and lacks cointegration test.

Response:

 We supplemented the relevant cointegration test results. The specific modifications are as follows:

 (1) The supplementary cointegration test results in section 3.1:

 Before performing impulse response function analysis and forecast error variance decomposition on the time series VAR model, it is necessary to conduct cointegration test on the model with the determined optimal lag period to judge whether the linear combination of stationary variable series with first-order lag has a stable equilibrium relationship, and ensure the effectiveness of VAR model estimation. The results of cointegration test are shown in Table 5.

Table 5. Results of the cointegration test at the world level.

 As can be seen from the estimation results in Table 5, the time series V AR model with first-order lag is stable, because the residual series of dRGI, d log(GDP) and d log(EXP) are all stable, i.e., there is a cointegration relationship between the variables.

 (2) The supplementary cointegration test results in section 3.2:

 When the 1st difference variables of RGI, log(GDP) and log(EXP) are stationary series, a cointegration test of each variable in the Panel VAR model is required to determine the validity of the model. The cointegration test results under different country categories are shown in Table 7.

Table 7. Results of the cointegration test at the country level.

 It can be seen from the test results of different groups that the probability p-values of the cointegration test statistics for each model are less than 0.01, i.e., they are all significant at the 1% significance level, and the original hypothesis is rejected. It shows that each variable in the Panel VAR model under different grouping samples has a cointegration relationship, which

means that the Panel VAR model analysis in the two cluster groups is effective.

 Please see section 3.1 and 3.2.

Suggestion 3:

 This paper only lists the empirical results and lacks in-depth analysis of the empirical results.

Response:

 We added in-depth analysis of the empirical results. The detailed additions are as follows:

 (1) Result analysis for Fig 1 as follows:

 ... ... The main reason is that reverse globalization has hindered the cross-regional flow of production factors such as resources, capital, technology, labor, gradually increased international trade tariffs and barriers, and the uncertainty of international political and social. The increase of international trade cost reduces the proportion of import and export trade and

foreign direct investment flows in which countries are involved, and ultimately inhibits export trade.

 (2) Supplements to the analysis of empirical results in section 3.2:

 1) Reason for Fig 2 and Fig 3 as follows:

 ... ... The possible reason is that, with the increasing strength of trade protectionism in developed countries such as Europe and the United States, countries rely less on cross-regional cooperation in trade in goods and services, encourage multinational enterprises and factories to return home to enhance the competitiveness of their own products, and their national export trade is less affected by reverse globalization. However, with the increasing uncertainty of international trade, reverse globalization has seriously hindered the opportunities for developing countries to participate in international economic and trade cooperation, restricted the spatial flow of various resources, capital, technology and other factors, and significantly increased trade costs, which in turn has a greater impact on the exports of developing countries.

 2) Reason for Fig 4 and Fig 5 as follows:

 ... ... The possible reason is that the EU, as the economic and political community of European countries, has a certain economic strength in its entire common market to resist the reverse globalization shocks. However, the participation of non-EU countries in multibilateral international trade cooperation is more vulnerable to factors such as tariffs, finance and international political instability. Since it is more costly for non-EU countries to participate in trade cooperation, the impact of reverse globalization on export trade is greater.

 (3) Reason for Table 11 as follows:

 ... ... This is principally because the secondary industrial exports are the main components of international trade, the manufacturing and construction industrial exports need a large number of production factors such as resources, capital and technology, reverse globalization increases the flow cost of various factors of production. Industries with high product

complexity often require more stringent technology and resource constraints, and countries have longer cycles of industrial restructuring, which eventually leads to the greater impact of reverse globalization on secondary industry exports.

 (4) Supplements to the analysis of empirical results in section 3.4:

 1) Reason for Fig 12 and Fig 13 as follows:

 ... ... The possible reason for this is that, China’s vast territory, large population, and abundant total resources make China as a major manufacturing exporter. In the development of reverse globalization, developed countries such as the United States encourage the return of multinational manufacturing enterprises and factories, which increases the unemployment pressure of Chinese residents and reduces China’s effective manufacturing output. The share of China’s manufacturing exports in international trade has decreased, which hinders the development of China’s manufacturing industry. As a result, the inhibitory effect of reverse globalization on China’s various sub-manufacturing exports is slightly greater than that of the United States.

 2) Reason for Fig 14 and Fig 15 as follows:

 ... ... The main reason is that the United States has obvious advantages in services exports. Its education, scientific and technological research, finance, insurance, communications and other industries have become the main force of US service industry exports, and the United States has extremely strong export competitiveness in technology-intensive service exports. Therefore, in the development of reverse globalization, the United States service industry exports are more sensitive to the impact of reverse globalization.

 Please see section 3.1, 3.2, 3.3 and 3.4.

Suggestion 4:

 This paper is lack of discussion section. It is suggested that the author compare the similarities and differences between the research results of this paper and the existing research results.

Response:

 We added the section of “4. Discussion” to compare the similarities and differences between the research results of this paper and the existing research results. The detailed modifications are as follows:

4. Discussion

 This paper empirically analyzes the dynamic impact of reverse globalization on export trade from the perspectives of world, country, industry, subdivided manufacturing and service industry. The study finds that reverse globalization has a significant inhibitory effect on export trade, which is similar to the results of Garg and Sushil [5], Dür et al. [8], He et al. [10], etc. The innovation of this paper is mainly from the perspective of country level and compares the differential impact of reverse globalization on the exports of different country categories. This paper holds that reverse globalization hinders the development of export trade in EU countries, which partially supports the research results of Dür et al. [8] and Li et al. [12]. In addition, this paper concludes for the first time that developing and non-EU countries’ exports are more affected by reverse globalization than those of developed countries and EU countries.

 This paper finds that reverse globalization has a significant negative impact on a country’s manufacturing exports, which confirms the research results of Garg and Sushil [5], Crino and Epifani [9], He et al. [10], etc. Furthermore, the innovation of this paper is mainly from the perspective of industrial and subdivided industry to analyze the heterogeneous effects of reverse

globalization on exports of different industry, subdivided manufacturing and service industry. The study finds that reverse globalization has the greatest inhibitory effect on the secondary industry exports, followed by the tertiary industry, and the inhibitory effect of reverse globalization on China’s subdivided industry exports is significantly greater than that of the United

States. Therefore, developing countries need to actively adjust their domestic industrial structure and export trade structure, enhance the export scale of industries with high technological complexity, and further increase trade cooperation with other countries.

 Please see section 4.

 Thanks again for Editors’and Reviewers’hard work. If there is any further improvement needed us to make, please do not hesitate to inform us.

---

## [Decision Letter · Decision Letter 1]

10 Jun 2022

Export trade structure transformation and countermeasures in the context of reverse globalization

PONE-D-21-35675R1

Dear Dr. Huang,

We’re pleased to inform you that your manuscript has been judged scientifically suitable for publication and will be formally accepted for publication once it meets all outstanding technical requirements.

Kind regards,

Ming Zhang, Ph.D.

Academic Editor

PLOS ONE

Additional Editor Comments (optional):

Reviewers' comments:

Reviewer's Responses to Questions

**Comments to the Author**

1. If the authors have adequately addressed your comments raised in a previous round of review and you feel that this manuscript is now acceptable for publication, you may indicate that here to bypass the “Comments to the Author” section, enter your conflict of interest statement in the “Confidential to Editor” section, and submit your "Accept" recommendation.

Reviewer #2: All comments have been addressed

2. Is the manuscript technically sound, and do the data support the conclusions?

Reviewer #2: Yes

3. Has the statistical analysis been performed appropriately and rigorously? 

Reviewer #2: Yes

4. Have the authors made all data underlying the findings in their manuscript fully available?

Reviewer #2: Yes

5. Is the manuscript presented in an intelligible fashion and written in standard English?

Reviewer #2: Yes

6. Review Comments to the Author

Reviewer #2: The authors revised this paper according to the comments of the reviewers, so it is recommended to accept it.

7. PLOS authors have the option to publish the peer review history of their article (what does this mean?). If published, this will include your full peer review and any attached files.

Reviewer #2: No

---

## [Editor Report · Acceptance letter]

15 Jun 2022

PONE-D-21-35675R1 

Export trade structure transformation and countermeasures in the context of reverse globalization 

Dear Dr. Huang:

I'm pleased to inform you that your manuscript has been deemed suitable for publication in PLOS ONE. Congratulations! Your manuscript is now with our production department. 

Kind regards, 

on behalf of

Dr. Ming Zhang 

Academic Editor

PLOS ONE